# THE ROLE OF PRE-TRAINING DATA IN TRANSFER LEARNING

## ABSTRACT

The transfer learning paradigm of model pre-training and subsequent fine-tuning produces high accuracy models. However, a question remains: what data and method should be used for pre-training? We study the effect of the pre-training data distribution on transfer learning in the context of image classification, investigating to what extent different pre-training datasets differ in downstream task performance. Through controlled experiments, we find that the pre-training dataset is initially important for low-shot transfer. However, the differences between distributions are diminished as more data is made available for fine-tuning. We also looked into how much is labeling worth compared to noisier but larger pre-training data. Our results show that to match the performance on supervised pretraining on ImageNet we need 15x-2000x more pre-train data from LAION for different downstream tasks. We also investigate the dataset size and observe that larger pre-training datasets lead to better accuracy, however, the absolute accuracy difference is the largest in the few-shot regime. Beyond data, we study the effect of the pre-training method, language-image contrastive vs. image-image contrastive, finding that the latter usually leads to better transfer accuracy.

## 1 INTRODUCTION

The best-performing computer vision models are produced by the transfer learning paradigm. In this two-step procedure, a model is first pre-trained on a large heterogeneous dataset. Next, the model is fine-tuned on application specific data which adapts the model to a problem of interest. While transfer learning is not new, it has become increasingly important with drastic improvements in the quality of pre-trained models (e.g., CLIP Radford et al. (2021), BASIC Pham et al. (2021), and Flamingo Alayrac et al. (2022)). These improvements are driven by new datasets for pre-training as well as better pre-training algorithms. This naturally leads to a question:

*How does the dataset and algorithm used for pre-training affect downstream transfer performance?*

While related works try to find the relation between pre-training and transfer performance by exploring scaling laws (Kornblith et al., 2019; Abnar et al., 2021) or predicting transferability without actual finetuning (You et al., 2021; Nguyen et al., 2020; Deshpande et al., 2021; Bolya et al., 2021), we highlight that to the best of our knowledge the role of pre-training data distribution has not been investigated so far. Therefore we define specific research questions detailed below and set up systematic experiments focusing on each question, while carefully ablating the other factors.

*To what extent do different pre-training datasets differ in downstream task performance?* Do we expect different distributions to perform differently in the transfer setting and how does that compare to training from scratch? When controlling for size but changing the pre-train dataset, we observe noticeable differences in downstream transfer accuracy. These differences are larger in the few-shot setting when only a few examples per class are available for fine-tuning. When many images are available for fine-tuning, the difference in absolute accuracy when varying the pre-training dataset mainly evaporates. Across many downstream tasks, certain pre-training datasets (i.e, Shutterstock) consistently lead to better transfer accuracy than others (i.e., WiT). However, there are still ordering differences from one downstream task to another. Moreover, even the pre-training dataset which leads to the worst transfer accuracy still outperforms training from scratch (see Figure 1).

*How much is expensive labeling worth compared to noisier but larger pre-training data?* We compare different pre-training strategies: supervised pre-training on small but labeled ImageNet and semi-supervised pre-training on image and language pairs on larger but noisier datasets. We find that pre-training on a well-curated dataset leads to better transfer accuracy than pre-training on a noisy dataset of similar size, and pre-training only on a 15x-2000x larger noisy dataset can close the gap (see Figure 2).

*How much does increasing pre-training dataset size contribute to the performance of transfer learning?* When controlling for the pre-training dataset and instead changing the size, we observe that models pre-trained on more images usually have better transfer performance. However, similarly to the aforementioned results, the absolute difference in transfer accuracy between models pre-trained on small scale and medium scale datasets is diminished when fine-tuning on more data. We also observe that increasing the pre-training dataset size shows different saturation performances on target tasks, i.e. while even 100X more data does not help on transfer to some downstream tasks, including more data improve the downstream performance of others (see Figure 3).

*What is the role of pre-training method on transfer performance?* We also examine the difference between supervised pre-training with the popular CLIP and SimCLR semi-supervised algorithms. Overall we find that the SimCLR pre-training leads to better transfer than CLIP pre-training in the low-shot regime, but that there are only small differences when many images are used for fine-tuning (see Figure 5).

To answer these questions we conduct an extensive empirical investigation (over 4000 experiments) in the context of computer vision. Our study covers 7 pre-training datasets (YFCC (Thomee et al., 2016), LAION (Schuhmann et al., 2021), Redcaps Desai et al. (2021), Conceptual captions-3m (Sharma et al., 2018) , Conceptual captions-12m (Changpinyo et al., 2021), WiT (Srinivasan et al., 2021), Shutterstock, ImageNet (Deng et al., 2009)), 9 fine-tuning datasets (CIFAR100 (Krizhevsky et al., 2009), DTD (Cimpoi et al., 2014), Caltech-101 (Fei-Fei et al., 2004), PETS (Parkhi et al., 2012), REAL and CLIPART from DomainNet (Peng et al., 2019), EuroSAT (Helber et al., 2019), Cassava Leaf Disease Classification (Cas), and Caltech Camera Traps-20 (Beery et al., 2018) ), and two pre-training methods CLIP (Radford et al., 2021) and SimCLR (Chen et al., 2020). To evaluate transfer performance, we examine both few-shot fine-tuning and full fine-tuning.

The paper is structured as follows: we review related work and provide relevant background on transfer learning in Section 2, followed by our experimental setup in Section 3. Section 4 details our observations relating to our research questions by measuring the downstream transfer accuracy models pre-trained on various data sources, dataset sizes, and with different pre-training losses. We discuss our findings and conclude with future research directions in Section 5.

## 2 RELATED WORK

Transfer learning is widely used in deep learning research and practice and has become a cornerstone in both computer vision and natural language processing. Through the years, there have been many questions on why transfer helps and how to choose a good pre-trained model to transfer from. Neyshabur et al. (2020) separated the effect of feature reuse from that of learning low-level pre-training data statistics. Their study involved models with supervised pre-trained on ImageNet. Another important question is whether transfer learning is always helpful on any downstream dataset. Raghu et al. (2019) experimented with downstream medical datasets (with images coming from a distribution that is very different from that of ImageNet or other natural image datasets) and found that transfer learning from ImageNet pre-trained models shows little benefit in performance. This shows that the downstream dataset is an important factor to consider when evaluating the transfer performance of upstream models. To make it possible to more generally evaluate the visual representations of upstream models, Zhai et al. (2019) introduced the Visual Task Adaptation Benchmark (VTAB). VTAB aims to measure the adaptability of representations to diverse, unseen tasks, given only a few examples from the downstream dataset.

Scaling up dataset and model size is a well-known trend for improving accuracy in both natural language processing (Kaplan et al., 2020) and computer vision (Kolesnikov et al., 2020). For instance, Kolesnikov et al. (2020) uses a weakly labeled JFT-300M dataset for pre-training. An even large noisy dataset with 3.5B images from Instagram was used in (Mahajan et al., 2018). To make use of large

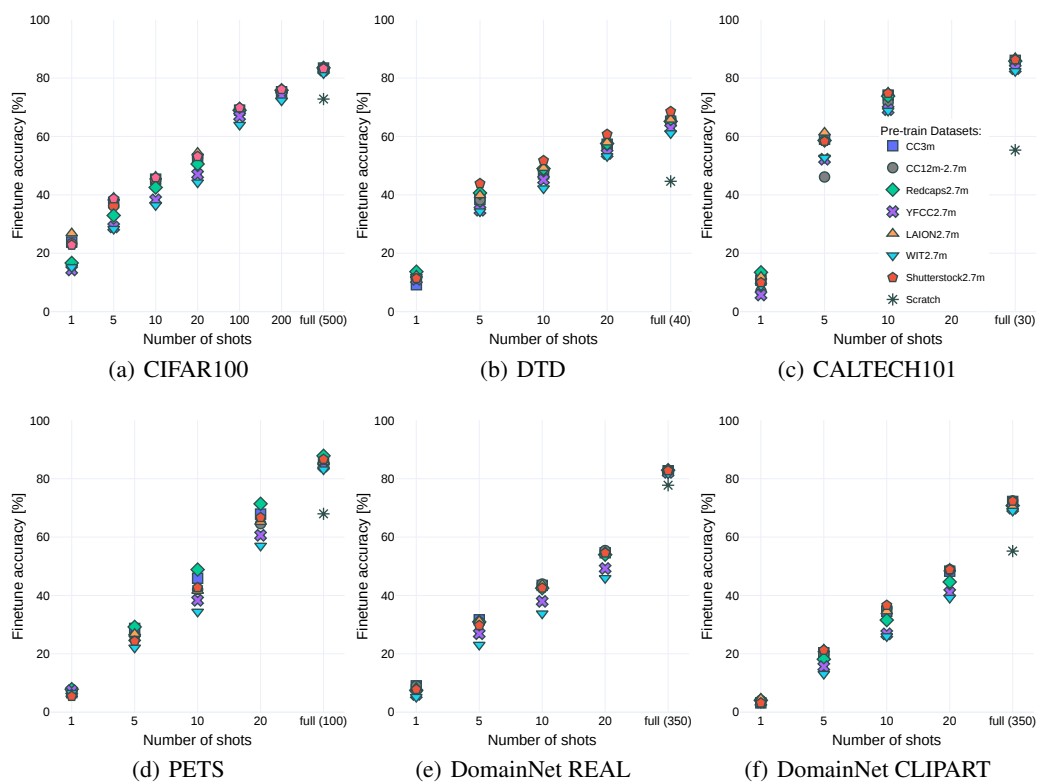

Figure 1: **Effect of the pre-training data distribution.** In the low-shot setting, different pre-training datasets lead to noticeable differences in downstream transfer performance. If many samples are available for fine-tuning, the difference in absolute accuracy between models pre-trained on different sources largely evaporates. See Figure 6 for an extension to more downstream datasets.

datasets without laborious labeling, researchers have proposed techniques to learn representations by unsupervised and self-supervised training. Chen et al. (2020); Caron et al. (2021); He et al. (2020); Grill et al. (2020) use image-only datasets to learn representations with self-supervised loss functions, one of the most popular being the contrastive loss. Ericsson et al. (2021) studied the transfer performance of self-supervised models and found that the best self-supervised models of that time could outperform supervised pre-training as an upstream source of knowledge transfer and that the performance of self-supervised models on ImageNet is indicative of downstream performance on natural image classification tasks. Similarly, Islam et al. (2021) found that contrastively trained models consistently outperform standard cross-entropy models in transfer learning. Goyal et al. (2021) showed that self-supervised models outperform supervised models on ImageNet, even when trained on random and uncurated images from the web. Moreover, they showed that these models are also good at few shot learning by achieving 77.9 % top-1 accuracy using only 10 % on ImageNet.

Building on contrastive techniques, Radford et al. (2021) introduced CLIP which learns a joint embedding space for both images and their descriptive captions, making it possible to effectively leverage a large-scale dataset from the Internet. Flamingo (Alayrac et al., 2022), a visual language model, is another successful example in the line of multimodal models and enables visual question answering and image captioning. CLIP and similar models like ALIGN (Jia et al., 2021), BA-SIC (Pham et al., 2021), and LiT (Zhai et al., 2022) demonstrated unprecedented robustness to challenging data distribution shifts. This accomplishment raised questions on the probable sources of such robustness—whether this robustness is caused by language supervision, the pre-training data distribution, size, or contrastive loss functions.

Fang et al. (2022) investigated this question and found that the diverse training distribution is the main cause of the robustness properties of CLIP. Nguyen et al. (2022) explored the role of the pre-training dataset for CLIP with a testbed of six pre-training sources, finding that no single pre-training dataset

consistently performs best. In recent work, Santurkar et al. (2022) carefully investigated the effect of language supervision in CLIP-like models, finding it an important factor if the pre-training dataset is large and the captions are descriptive enough. Unlike their work, we consider end-to-end fine-tuning which result in higher accuracy.

Kim et al. (2022) conducted an in-depth study of the effect of the network architecture, pre-training dataset, supervised vs self-supervised learning objectives, and different domain transfer methods on the transferability of representations to new domains. They found that the transferability of the pre-trained representations depends on factors such as the target benchmark, adaptation method, and network depth.

Our work is closely related to Abnar et al. (2021) where the authors explored how different upstream training settings affect the upstream and downstream accuracy for two upstream datasets and more than 20 downstream tasks. They showed that as the upstream accuracy increases, the transfer learning performance on downstream datasets saturates. However, the authors study only upstream models that are pre-trained with a supervised loss function on ImageNet-21K (Deng et al., 2009) or JFT-300M (Sun et al., 2017). In this work, we extend these results to more pre-training datasets and methods. Moreover, we consider full fine-tuning in addition to few-shot transfer.

## 3 EXPERIMENTAL SETUP

**Model**. The main focus of this study is the CLIP model (Radford et al., 2021). This model has demonstrated unprecedented robustness to natural distribution shifts (Taori et al., 2020; Miller et al., 2021), and transfers well to many downstream tasks (Radford et al., 2021; Wortsman et al., 2021). Given an image-text pair, CLIP learns a joint embedding space for both images and their captions and tries to maximize the cosine similarity between the text and image embedding for an image relative to the cosine similarity of unaligned pairs. We use the CLIP implementation from the OpenCLIP GitHub repository (Ilharco et al., 2021).

**Pre-training**. We mainly use ResNet-50 (He et al., 2016) as the image encoder unless stated otherwise. We vary the pre-training data distribution, curation method in Section 4.1, and pre-training dataset size in Section 4.2 to obtain different pre-trained models. We also change the contrastive loss function to SimCLR in Section 4.3 to test the effect of the pre-training method on downstream transfer accuracy. Further training details are in Appendix A.

**Fine-tuning**. For most of the experiments we finetune the pretrained model end-to-end on the target transfer dataset unless otherwise stated. For each pre-trained model and downstream transfer dataset, we used a large grid search over various fine-tuning hyperparameters including learning rate, batch size, and the number of epochs. We report the best-performing accuracy in the plots. Further training details are in Appendix A.

**Datasets**. Our large-scale experiments yield more than 1000 trained networks. These experiments consist of pre-training on 7 datasets and fine-tuning on 6 downstream tasks. Our pre-training datasets include:

- YFCC: Our experiments mostly include YFCC-2.7M, a random subset of YFCC-15M. The 15M subset of the YFCC-100M dataset (Thomee et al., 2016) was filtered to only include images with English titles or descriptions. The dataset contains 14,829,396 images with natural language captions associated with each image. The images and captions are collected from Flickr.

- LAION (Schuhmann et al., 2021): The images and corresponding alt-texts come from web pages collected by Common Crawl (Com) between 2014 and 2021. We randomly select a subset of 2.7M and 15M samples for our experiments.

- Redcaps Desai et al. (2021): Redcaps contains 11,882,403 examples from 350 manually curated subreddit collected between 2008 and 2020. The subreddits are selected to contain a large number of image posts that are mostly photographs and not images of people.

- Conceptual Captions-3m (Sharma et al., 2018): The raw descriptions in Conceptual Captions are harvested from the alt-text HTML attribute associated with web images. This dataset contains 2,799,553 samples, denoted as CC_2.7m in the plots.

- Conceptual Captions-12m (Changpinyo et al., 2021): A dataset with 12 million image-text pairs. It is larger than CC_2.7m and covers a much more diverse set of visual concepts. We randomly select 2.7M samples from this dataset, denoted as CC_12_2.7m.

- WIT (Srinivasan et al., 2021): Image-text pairs come from Wikipedia pages. We use reference description as the source of text data and obtain 5,038,295 examples in total after filtering to include only the English language.

- Shutterstock: 11,800,000 images and captions were crawled from the Shutterstock website in 2021. We randomly sample this dataset.

For downstream tasks we use the nine different datasets (CIFAR100 (Krizhevsky et al., 2009), DTD (Cimpoi et al., 2014), Caltech-101 (Fei-Fei et al., 2004), PETS (Parkhi et al., 2012), REAL and CLIPART from DomainNet (Peng et al., 2019), EuroSAT (Helber et al., 2019), Cassava Leaf Disease Classification (Cas), and Caltech Camera Traps-20 (Beery et al., 2018)). See Appendix Section B and Table 3 for details on target transfer datasets.

## 4 EXPERIMENTS

### 4.1 EFFECT OF PERTAINING DATA SOURCE

In this section, we study the role of the pre-training data distribution. We do so by first pre-training on various noisy pre-training data sources and comparing downstream transfer accuracy. Next, we test the effect of pre-training on the curated dataset ImageNet Deng et al. (2009).

**What is the role of the pre-training data source in transfer learning?** For a controlled experiment to study the effect of the dataset on the downstream transfer accuracy, we fix the learning algorithm to CLIP, architecture to ResNet-50 He et al. (2016), and the number of images to 2.7 million. We get 2.7 million images by randomly subsampling the original datasets. Figure 1 compares different data sources for pre-training. We make the following observations in Figure 1 on the effect of the pre-training data distribution on the downstream transfer accuracy:

(1) Changing the pre-training dataset leads to noticeable differences in downstream transfer performance. However, these differences are most pronounced in the low-shot setting. When many images are available for fine-tuning, the difference in absolute accuracy between different pre-training models is largely diminished. For CIFAR100, REAL, and CLIPART the model fine-tuned on the full dataset has very similar transfer performance despite different pre-training datasets. While this is not the case for DTD, CALTECH101, and PETS, these datasets have far fewer images per class available when fine-tuning on the full dataset for the downstream task, hence the difference between different pre-train datasets is still noticeable.

(2) We also notice that the ordering of models with respect to their performance on the downstream datasets is largely consistent with Shutterstock and LAION being the best performing pre-training datasets for different downstream tasks. WIT yields the worst performance in most cases.

(3) While transfer learning from a large pre-training dataset outperforms training from scratch for all downstream tasks, the magnitude of the improvement varies for different datasets. For example, we observe a large improvement for PETS and CLIPART and a smaller effect of pre-training for REAL.

**Do well-curated pre-training datasets lead to better transfer?** There has been a significant effort to create computer vision datasets with high-quality images and labels. On the other hand, many recent datasets are large but noisy. In this set of experiments, we are going to answer two specific questions: *How much is ImageNet labeling worth? Is there anything special about ImageNet data distribution?*

In Figure 2 we first start by pre-training ResNet-50 on ImageNet-1K using supervised cross-entropy loss and finetune on the set of our target tasks as a baseline. While we want to see the effect of curation, we then discard ImageNet labels and use CLIP to pre-train on ImageNet. Because the ImageNet dataset has no captions, we include original Flickr captions, which reduces the size of the image and captions to 0.5M samples (more details in Appendix section C). We observe that supervised pre-training on ImageNet outperforms CLIP pretraining on ImageNet with Flickr captions

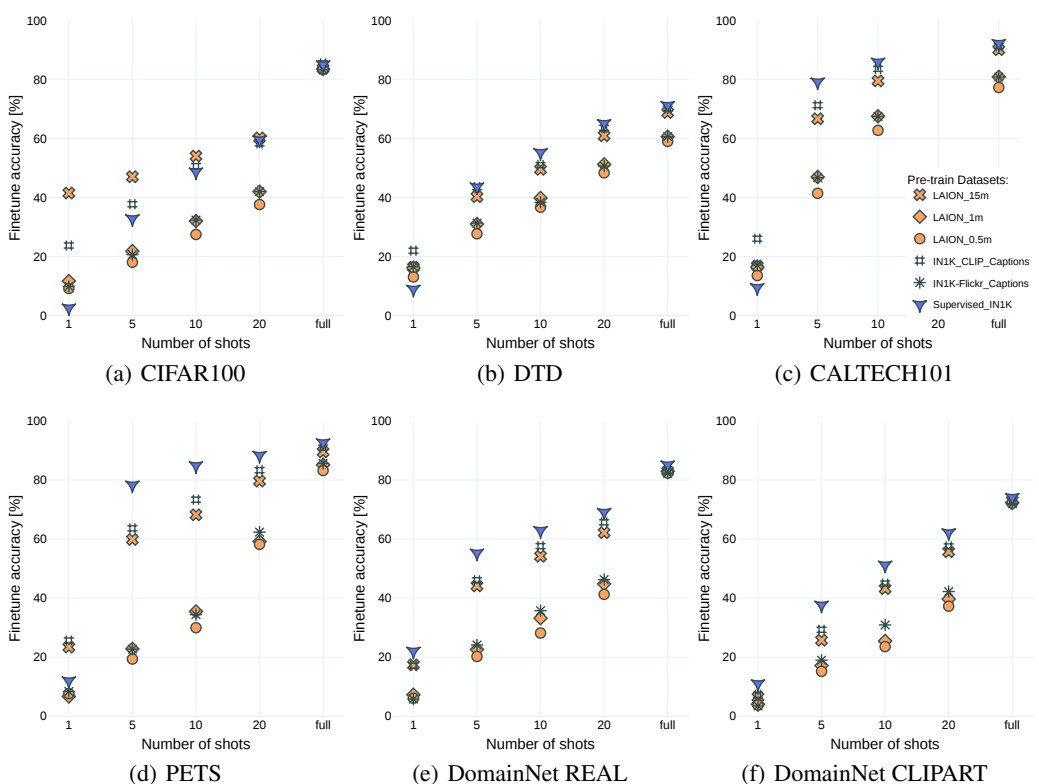

Figure 2: **Effect of data curation.** Pre-training on a curated dataset leads to better transfer accuracy than pre-training on a noisy dataset of similar size. However, pre-training on a noisy dataset that is 15x larger overall performs similarly to pre-training on a curated dataset, especially in the high-shot transfer regime.

by a large margin in all target tasks. We then use all the images from ImageNet paired with templated captions, e.g., "a photo of a class name". This allows us to have a fair comparison between supervised and CLIP pre-training on ImageNet, given the same size. We observe that pre-training with new captions improves the performance of CLIP pre-training by a large margin, defeating supervised pre-training on CIFAR100. However, supervised pre-training on ImageNet still performs best for the rest of the five datasets.

we also compare ImageNet distribution with LAION in Figure 2. Pre-training CLIP on LAION-1m performs similarly to ImageNet with Flickr captions(0.5m). Interestingly for contrastive CLIP pre-training, we need 15x more data from LAION to match the performance on ImageNet with CLIP captions.

We also train CLIP models on subsampled versions of the large noisy LAION dataset of the same size 0.5m and 1m. Moreover, we test the effect of pre-training on a lot of noisy data by also pre-training on 15m samples from LAION.

Figure 2 highlights the impact of *data curation* on the quality of the pre-trained models with respect to their transfer accuracy on downstream tasks. We observe:

(1) If we fix the number of samples to 0.5m, LAION-0.5m shows the worst performance for all downstream tasks. Comparing LAION-0.5m with IN1K-Flickr-Captions shows that pre-training on clean, curated images and their associated (possibly noisy) captions improves transfer accuracy.

(2) Pre-training a CLIP model on a 2 times larger yet noisy dataset LAION-1m can outperform the IN1K-Flickr-Captions or yield very similar performance.

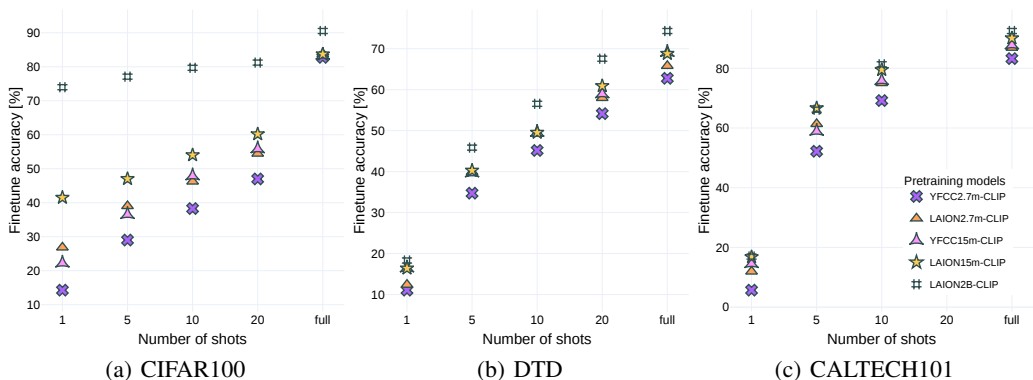

Figure 3: **Effect of the pre-training dataset size.** Increasing the size of the dataset used for pre-training results in better transfer accuracy on downstream tasks. However, the absolute accuracy difference is smaller in the high-shot regime, even when pre-training consists of $100\times$ more data. Still, using an extremely large pre-training dataset of LAION-2B results in significantly better accuracy on CIFAR100 and DTD.

(3) Next we study the effect of clean images and captions by comparing LAION-1m with IN1K-CLIP-Captions. We observe that IN1K-CLIP-Captions results in better downstream transfer accuracy on all transfer datasets.

(4) However, pre-training a CLIP model on a 15 times larger yet noisy dataset LAION-15m can outperform the IN1K-Flickr-Captions on CIFAR100, and yield very similar performance on the remainder of the datasets.

(5) Similar to the findings in Figure 1, we observe more noticeable differences in terms of the absolute accuracy difference in the low-shot regime. In the high-shot regime, the difference in absolute accuracy between the different pre-training datasets is smaller.

## 4.2 EFFECT OF PRE-TRAINING DATASET SIZE

To study the effect of the pre-training dataset size on downstream transfer accuracy, we investigate different subsample sizes for the YFCC and LAION datasets. Specifically, we fix the pre-training distribution to YFCC and LAION and compare pre-training on 2.7m samples with 15m samples. We also extended our experiments to see the effect of extreme sample sizes and include ViT-B/32 CLIP model trained on 2B samples from LAION. The results are presented in Figure 3. We make the following observations:

(1) Increasing the size of the dataset used for pre-training results in higher downstream transfer accuracy. However, the magnitude of the improvement varies across different downstream datasets. Increasing the size of the datasets subsampled from YFCC and LAION improves the downstream performance on CIFAR100 by a large margin. However, the improvement is more modest for DTD and CALTECH101.

(2) Similarly to the findings in Figure 1, we observe more noticeable differences in downstream transfer performance in the few-shot regime when fixing the dataset and varying the number of samples. The difference in the absolute accuracy when more data is available for fine-tuning is usually smaller.

(3) LAION pre-training outperforms YFCC pre-training on downstream accuracy on CIFAR100, DTD, and CALTECH—Interestingly on these tasks LAION-2.7m performs similarly to a much larger size of YFCC-15m pre-training.

(4) Using an extremely large dataset of LAION-2B improves the performance by a significant margin in the low-shot regime for CIFAR100. While differences in absolute accuracy are less when more data is available for fine-tuning, LAION-2B pre-training still performs consistently better. Moreover, the differences on DTD and CALTECH101 are overall much smaller.

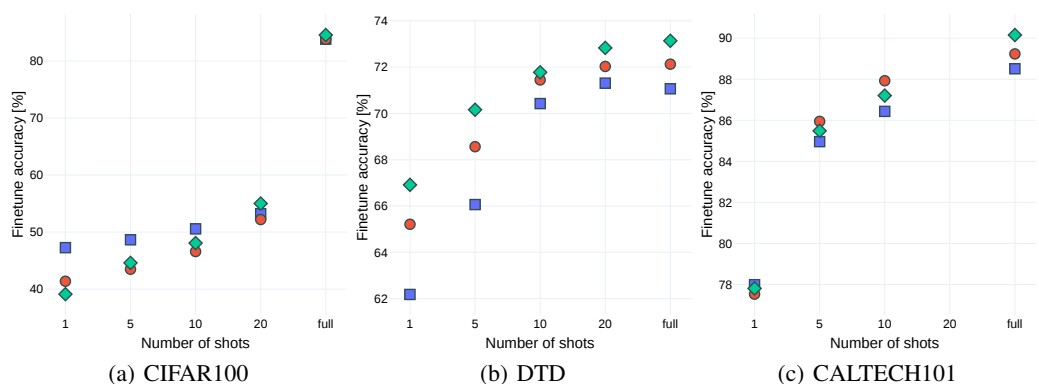

(a) CIFAR100         (b) DTD         (c) CALTECH101

Figure 4: **Effect of the pre-training data distribution when using SimCLR as the pre-training method.** Using different datasets for pre-training leads to noticeable difference in downstream transfer accuracy. Similar to the previous results for CLIP pre-training, the absolute difference in downstream transfer accuracy between different pre-training datasets is smaller when many images are available for fine-tuning.

## 4.3 EFFECT OF PERTAINING LOSS

In this section, we investigate the effect of the method used for pre-training on downstream transfer accuracy. Therefore, we change the pre-training loss from language-image contrastive (CLIP Radford et al. (2021)) to image-only contrastive (we use SimCLR (Chen et al., 2020)). In contrast to previous experiments with CLIP where we fine-tuned end-to-end from the zero-shot pre-trained model, here we fine-tune using LP-FT (Kumar et al., 2022). We do this because we are no longer able to start with a zero-shot pre-trained model. When we compare to CLIP, we fine-tune both models with LP-FT to facilitate a fair comparison.

LP-FT is the following two-step procedure: for each number of shots $k$ we first freeze the encoder and train a classification head from random initialization using $k$ examples per-class from the downstream task. In the second step, we initialize the classification head with this linear probe (LP) then unfreeze all weights and finetune (FT) the whole model.

Figure 4 shows the results of changing the pre-training dataset when SimCLR is used as the pre-training method. We make the following observations:

(1) Similarly to Figure 1, changing the pre-train dataset leads to differences in the downstream transfer performance. However, the differences in absolute accuracy are smaller when many images are available for fine-tuning.

(2) We previously observed that for CLIP, Shutterstock led to better transfer accuracy in most settings. However when pre-training with SimCLR, the Redcaps datasets results in higher downstream transfer accuracy.

Next, we test the difference between CLIP and SimCLR pre-training. The results are presented in Figure 5. For a fair comparison, we finetune both CLIP and SimCLR models using LP-FT (Kumar et al., 2022). We summarize our findings below:

(1) Overall we find that models pre-trained with SimCLR have better downstream transfer accuracy than models pre-trained with CLIP in the low-shot regime.

(2) Similarly to our observations regarding the effect of the pre-training data distribution (Figures 1 and 4), the absolute accuracy difference is smaller when more data is used for fine-tuning. We note that this is different from what was observed by Santurkar et al. (2022). However, we suspect this difference is because we are fine-tuning all model parameters while they consider only a linear classifier. We are interested in modifying all parameters as this results in higher accuracy.

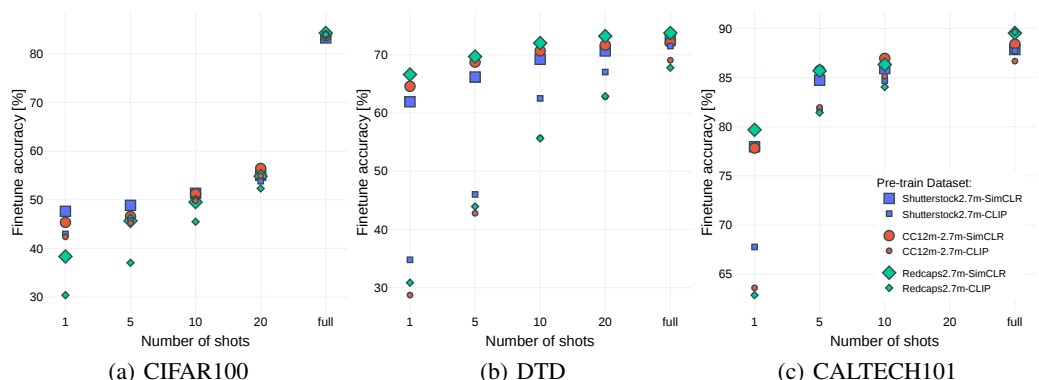

Figure 5: **CLIP vs. SimCLR for pre-training.** Overall we observe that SimCLR pre-training leads to better downstream transfer accuracy than using CLIP pre-training on the same dataset. These differences are most substantial in the few-shot setting.

(3) The difference in downstream transfer accuracy for CLIP and SimCLR pre-training varies across different datasets. While SimCLR is only marginally better than CLIP for CIFAR100, the difference is significantly larger for DTD and CALTECH101, especially in the low-shot setting.

## 5 DISCUSSION, LIMITATIONS, AND FUTURE WORK

**Discussion.** As better pre-trained models become available, and more workloads shift from training from scratch to fine-tuning, understanding the transfer learning paradigm becomes increasingly important. Presumably, in the future, a sea of pre-trained models will be available for download from the Internet. Therefore, researchers and practitioners will be faced with the question of where to begin. It will be important to make this choice well, but also to understand to what extent this choice matters.

Overall we have observed that different pre-training distributions and methods can lead to differences in downstream transfer accuracy. However, these differences are largest in the few-shot transfer regime, and once many images are used for fine-tuning these differences are mostly diminished. Moreover, while different pre-training decisions lead to similar accuracy in the high-shot regime, they still outperform training from scratch in the setting we consider.

**Limitations and Future work** There are a number of limitations in our study. For one, we consider only end-to-end fine-tuning. We choose this because it is the method of fine-tuning that produces the highest accuracy. However, if compute is limited one may choose to instead use only a linear probe or other more lightweight methods of fine-tuning. So far this is not addressed in our study.

Another limitation is that we do not do an exhaustive hyperparameter sweep for pre-training. While fine-tuning is cheaper and we are therefore able to do a grid search, for pre-training we are mostly limited to using existing checkpoints. While we feel as though this reflects a realistic setting, in future work our goal is to better understand the role of hyperparameters.

In addition to mentioned limitations, future works might include understanding why supervised pre-training on smaller but well-curated ImageNet shows superior performance. The main question is what is special about ImageNet-1K and studied downstream tasks.

## 6 REPRODUCIBILITY AND ETHICS STATEMENS

We strongly believe in the reproducibility of our results and therefore included the details of used datasets in the section 3. We also included the details of the pre-training and fine-tuning in section 3 and Appendix section A.

We believe that our submission raises no questions regarding the Code of Ethics as we use open public datasets and implementations. Our study also does not involve human subjects, potentially harmful insights, methodologies and applications, potential conflicts of interest and sponsorship, discrimination/bias/fairness concerns, privacy and security issues, legal compliance, and research integrity issues. We also referred to close related works and compare our results wherever applicable.

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

APPENDIX

# A  TRAINING DETAILS

## A.1  CLIP TRAINING

Our CLIP models are trained from scratch on each of the pre-training datasets unless otherwise mentioned and follow the training code from the OpenCLIP GitHub repository(Ilharco et al., 2021). CLIP models are trained using AdamW optimizer (Loshchilov & Hutter, 2017) with default PyTorch parameters $\beta_1 = 0.9$, $\beta_2 = 0.999$, $\epsilon = 10^{-8}$, batch size 1024, and weight decay of 0.1. For learning rate, we start with a learning rate of $10^{-3}$ and apply a cosine-annealing learning rate schedule (Loshchilov & Hutter, 2016) with 5,000 steps warm-up. We use the same data augmentations as in(Radford et al., 2021).

## A.2  SIMCLR TRAINING

Our SimCLR implementation closely follows the training code from the SLIP(Mu et al., 2021). SimCLR models are also trained for 16 epochs from scratch using AdamW optimizer (Loshchilov & Hutter, 2017) with $\beta_1 = 0.9$, $\beta_2 = 0.98$, $\epsilon = 10^{-8}$, batch size 1024, and weight decay of 0.1. we start with a learning rate of $10^{-3}$ and apply a cosine-annealing learning rate schedule (Loshchilov & Hutter, 2016) with 2 epochs of warm-up. The hidden dimension of SimCLR MLP projection head is set to 4,094 and the output embedding dimension of MLP projection head is set to 256.

## A.3  FINETUNING DETAIL

Each pretrained model is finetuned on the specific downstream task for 128 epochs while the learning rate is from 0.0001, 0.0003, 0.001, 0.003 as starting and applying a cosine-annealing learning rate schedule (Loshchilov & Hutter, 2016) with 500 steps warm-up and batch size of 128. For each fine-tuning, we choose the best performing result on the test set among the performed grid search. We use the implementation from the WiSE-FT GitHub repository for fine-tuning, where we have only one model and $\alpha = 1$ (Wortsman et al., 2021).

# B  EFFECT OF THE PRE-TRAINING DATA DISTRIBUTION

Here we extend the experiments in section 4.1 to include three more downstream datasets. While the first six datasets in Figure 1 are internet-crawled datasets, these three new downstream datasets are domain-specific, *i.e.*, the dataset is created after a specific challenge is defined in a specific domain.

- EuroSAT (Helber et al., 2019): The task is to classify land use and land cover based on Sentinel-2 satellite images. The dataset covers 13 spectral bands and consists of 10 classes within total of 27,019 labeled and geo-referenced images. we create an $80\%$-$20\%$ random class-balanced split with the provided dataset.

- Cassava Leaf Disease Classification (Cas): The dataset contains 21,397 images from the Kaggle competition, to give farmers access to methods for diagnosing plant diseases. The images are labeled as healthy or as one of four different diseases. we split the dataset with $80\%$-$20\%$ random class-balanced ratio for train and test, respectively.

- Caltech Camera Traps-20 (Beery et al., 2018): CCT-20 contains 57,864 images in 15 classes, taken from camera traps deployed to monitor animal populations. Classes are either single species *e.g.*, "Coyote") or groups of species, *e.g.*, "Bird"). CCT-20 is a subset of the iWildCam Challenge 2018, whose yearly editions have been hosted on Kaggle. Here we study the subset of CCT-20 that come from the same locations [1], including 14,071 and 16,395 images for train and test respectively.

Figure 6 compares different data sources for pre-training. While Shutterstock shows superior performance on the first six datasets (except for PETS), the best pre-training distribution changes

---

[1]"Cis" in the main dataset refers to images from locations seen during training, and "trans" refers to new locations not seen during training

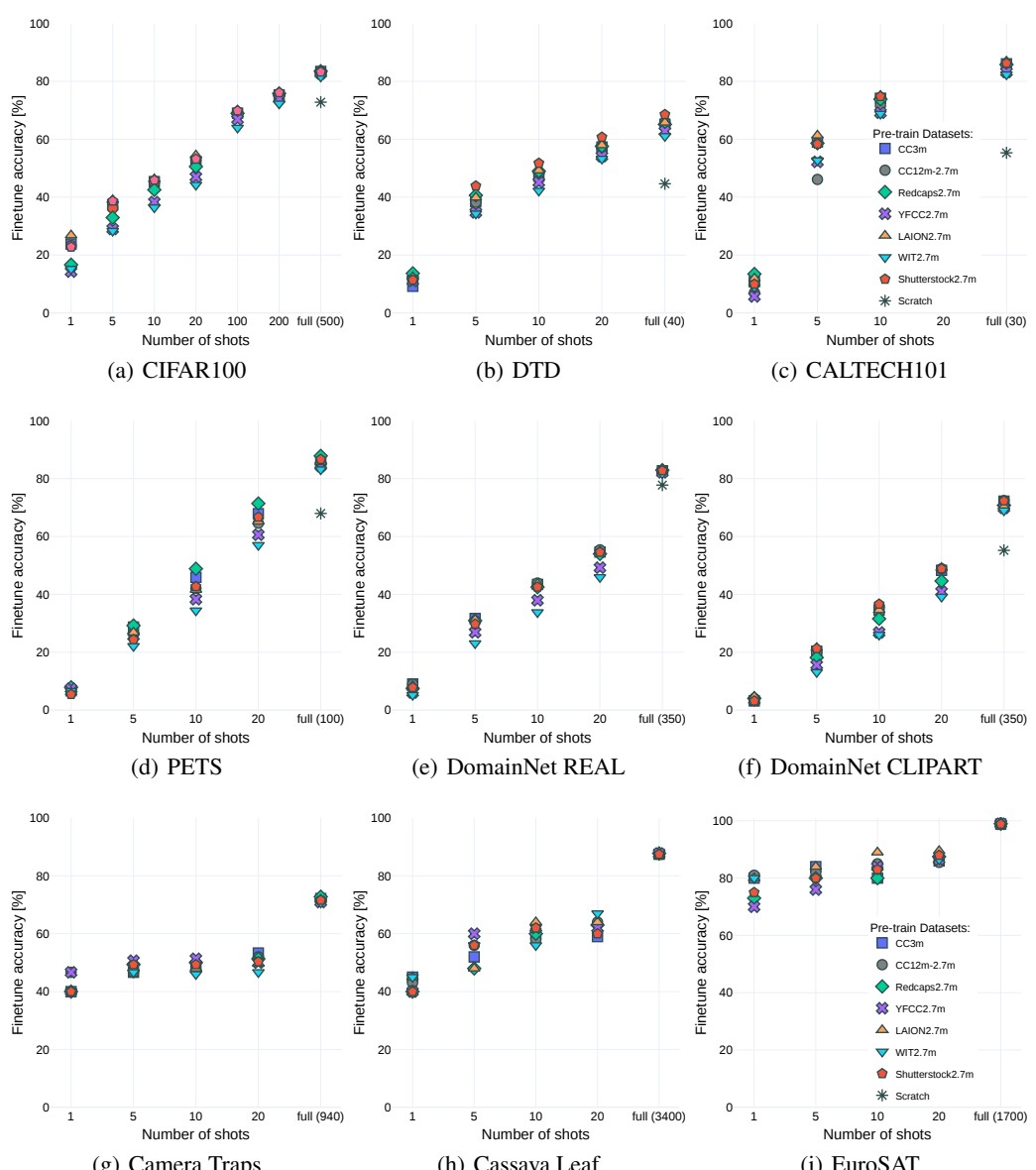

Figure 6: **Effect of the pre-training data distribution.** We extend Figure 1 to include three more downstream datasets of Camera Traps, Cassava Leaf, and EuroSAT. In the low-shot setting, different pre-training datasets lead to noticeable differences in downstream transfer performance. If many samples are available for fine-tuning, the difference in absolute accuracy between models pre-trained on different sources largely evaporates.

between Camera Traps, Cassava Leaf, and EuroSAT. Changing the pre-training dataset leads to noticeable differences in the downstream low-shot transfer performance of nine datasets.

## C  EFFECT OF DATA CURATION: IMAGENET CAPTIONING

We compare CLIP models pre-trained on LAION with CLIP models pre-trained on the following two versions of the curated ImageNet dataset:

- IN1K-Flickr-Captions: This is a subset of the ImageNet Large Scale Visual Recognition Challenge (ILSVRC) 2012 training set, paired with the original image title, description, and

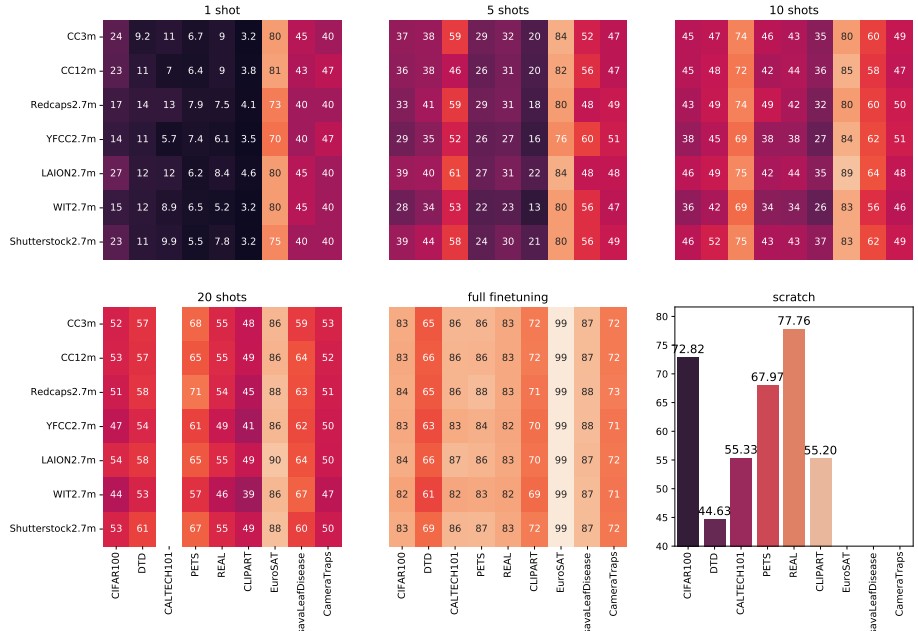

Figure 7: **Effect of pre-training data distribution: a better view.** We change the presentation of Figure 1 for a better view of exact performance numbers on different data distributions and datasets.

tags from Flickr. Therefore, we can use it for CLIP pre-training. To construct this dataset, Fang et al. (2022) start from 14,197,122 image URLs in the ImageNet fall 2011 release, and filter to only include images from Flickr. Next, they restrict the images to the 1,000 classes included in the 2012 ImageNet competition, run the image deduplication routine, and remove text containing profanity. As a result, the dataset of 463,622 images is left along with the newly obtained corresponding text data.

• IN1K-CLIP-Captions: This dataset includes all data in the ImageNet dataset, paired with templated captions, e.g., "a photo of a classname". This allows us to use CLIP pre-training but on clean images and text. In terms of ImageNet accuracy, this training scheme is very similar to standard supervised training. However, this is now a controlled experiment as we are always using CLIP pre-training.

# D  EFFECT OF THE PRE-TRAINING DATASET SIZE

in Figure 8 we extend the experiments in 4.2 to include three more downstream datasets of PETS, REAL, and CLIPART. Similar to Figure 3 we observe that increasing the size of the dataset used for pre-training results in higher downstream transfer accuracy. However, the magnitude of the improvement varies across different downstream datasets. While increasing the size of the datasets improves the downstream performance on CLIPART(and CIFAR100) by a large margin, the improvement is more modest for PETS and REAL. Similar to Figure 3, we also observe that LAION pre-training outperforms YFCC pre-training on downstream accuracy. On these tasks, LAION-2.7m performs similarly to a much larger size of YFCC-15m pre-training.

# E  OTHER ARCHITECTURES

In order to see the effect of architecture on the observed trends, we extend the results to the effect of pre-training distribution in Figure 1 to include Vison Transformers. To do so, we used ViT-B/32 released checkpoints trained on LAION-400m and OpenAI-400m, . Figure 9 shows the effect of data distribution on finetune transfer to CIFAR100, DTD, and CALTECH101 when using ViT instead of

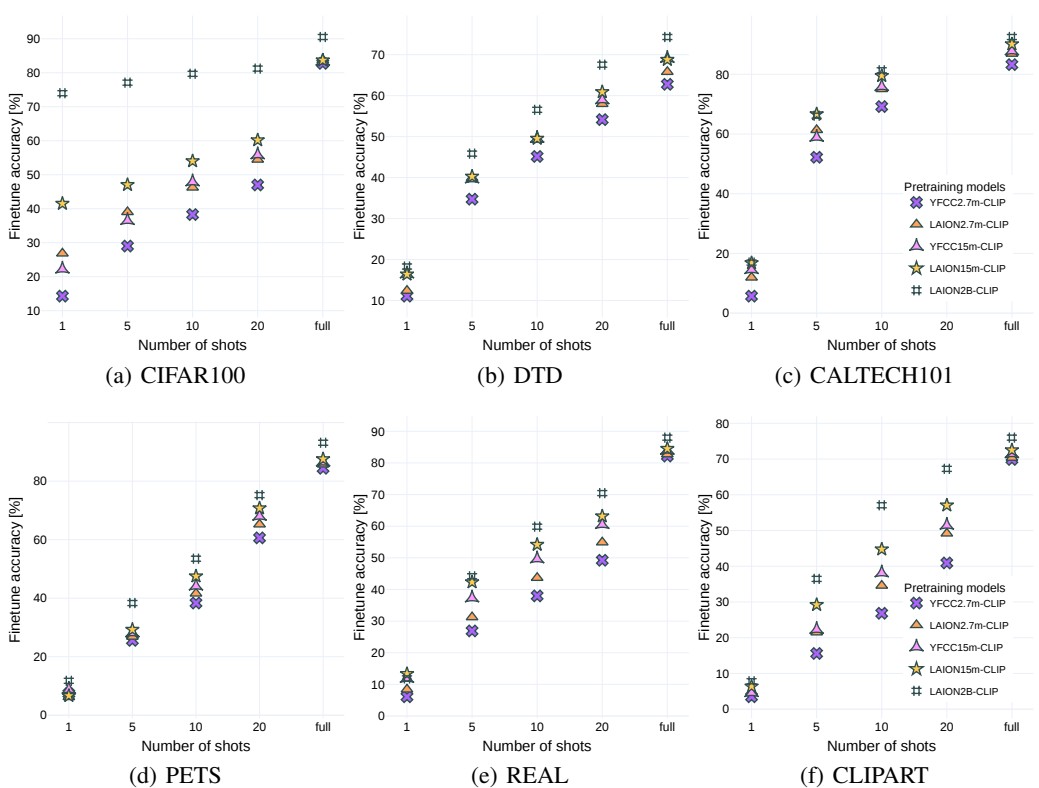

Figure 8: **Effect of the pre-training dataset size.** Increasing the size of the dataset used for pre-training results in better transfer accuracy on downstream tasks. However, the absolute accuracy difference is smaller in the high-shot regime, even when pre-training consists of $100\times$ more data. The benefit of pre-training on LAION-2B is different on target tasks. While there is major gap between LAION-2B and LAION-15m for CIFAR100, the performance on CALTECH101 is saturated.

ResNet-50. While similar to Figure 1 the difference between the fine-tune performance is minimal, we observe that both models perform also very similar in the few-shot setting. We hypothesize that this observation could be attributed to the similarity between LAION and OpenAI distributions rather than employing transformer instead of ResNet-50. A controlled study may include to replicate Figure 1 but with ViT, and we leave that for future work.

## F    EFFECT OF PRE-TRAINING DATA DISTRIBUTION: SIMCLR INSTEAD OF CLIP

We extend the set of downstream task from three datasets in Figure 4 to six in Figure 10. Findings in Figure 4 still hold true for other downstream tasks. We also observe that similar to using CLIP, Redcaps using SimCLR shows superior performance on PETS.

## G    DATASETS

### G.1    PRE-TRAINING DATASETS

Our study covers 7 pre-training datasets including (YFCC (Thomee et al., 2016), LAION (Schuhmann et al., 2021), Redcaps Desai et al. (2021), Conceptual captions-3m (Sharma et al., 2018) , Conceptual captions-12m (Changpinyo et al., 2021), Shutterstock, ImageNet-Captions (Fang et al., 2022)). Table 1 shows their main source and total size. We also show some examples of image-caption pairs randomly selected from Shutterstock in Figure 11, Redcaps in Figure 12, YFCC-15m in  Figure 13,

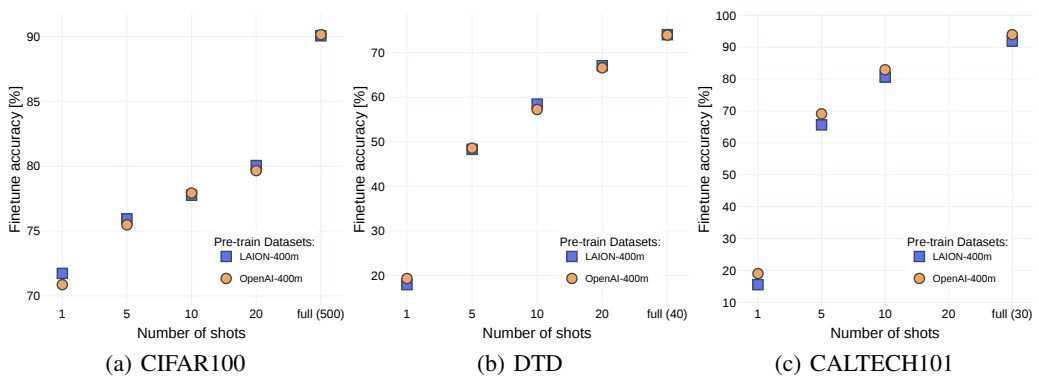

(a) CIFAR100  (b) DTD  (c) CALTECH101

Figure 9: **Effect of the pre-training data distribution: ViT instead of ResNet-50** While similar to Figure 1 the difference between the fine-tune performance is minimal, we observe that both models perform also very similarly in the few-shot setting. We hypothesize that this observation could be attributed to the similarity between LAION and OpenAI distributions rather than employing transformer instead of ResNet-50.

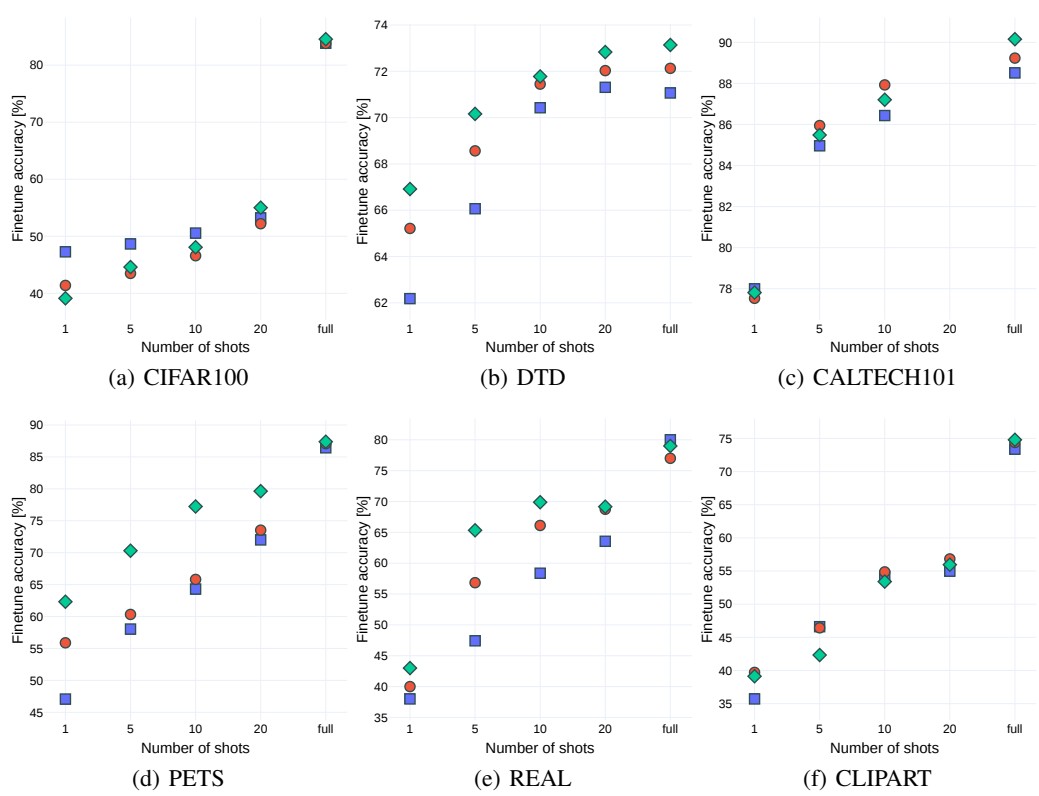

(a) CIFAR100  (b) DTD  (c) CALTECH101

(d) PETS  (e) REAL  (f) CLIPART

Figure 10: **Effect of the pre-training data distribution when using SimCLR as the pre-training method.** Using different datasets for pre-training leads to noticeable difference in downstream transfer accuracy. Similar to the previous results for CLIP pre-training, the absolute difference in downstream transfer accuracy between different pre-training datasets is smaller when many images are available for fine-tuning.

LAION-15m in Figure 14, Conceptual Captions in Figure 15, and WIT in Figure 16. Table 2 also shows the most common words in captions of these pre-training datasets.

Looking at Redcaps samples in Figure 12 and also the top 20 captions shows many samples of animals. This is showing why Redcaps perform better on PETS. Samples from WIT in Figure 16 and also its top 20 words mostly featuring geographical locations, which is rare in our downstream task, hence performing worst compared to other pre-training distributions. Shutterstock top 20 words also include words like "pattern", "texture", "and design" which are close to DTD classes, hence showing superior performance in this downstream task.

| Dataset | Source | Total size |
|---------|--------|-----------|
| YFCC | Flickr | 14,826,000 |
| LAION | Common Crawl | 15,504,742 |
| CC-12M | Unspecified web pages | 9,594,338 |
| RedCaps | Reddit | 11,882,403 |
| WIT | Wikipedia | 5,038,295 |
| ShutterStock | ShutterStock | 11,800,000 |
| IN1K-Captions | ImageNet | 463,622 |

Table 1: Details on pre-training datasets

| Pre-training dataset | Top 20 words in 1M sample of captions |
|----------------------|----------------------------------------|
| Shutterstock | **background**, vector, illustration, **design**, icon, **pattern**, **texture**, style, woman, concept, hand, color, flower, view, template, line, business, logo, card, symbol |
| Redcaps | day, today, year, time, cat, plant, friend, anyone, picture, baby, guy, week, dog, home, morning, night, month, way, boy, work |
| YFCC-15m | photo, day, park, street, city, picture, view, time, world, year, house, state, center, part, garden, shot, image, building, road, museum |
| LAION-15m | photo, stock, image, black, woman, design, set, vector, white, print, home, men, blue, dress, art, card, sale, gold, bag, cover |
| CC-12m | illustration, stock, art, design, photo, image, background, room, vector, house, home, woman, wedding, style, photography, royalty, car, fashion, girl, world |
| CC-3m | background, actor, artist, player, illustration, view, woman, man, football, team, tree, premiere, city, vector, day, girl, beach, game, hand, people |
| WIT | view, church, station, map, house, building, hall, museum, city, location, street, park, river, state, john, county, town, center, bridge, world |

Table 2: Most common words in captions of pre-training distributions

## G.2 DOWNSTREAM TASKS

We measure the transferability of learned representations using different methods on CIFAR100 (Krizhevsky et al., 2009), DTD (Cimpoi et al., 2014), Caltech-101 (Fei-Fei et al., 2004), PETS (Parkhi et al., 2012), REAL and CLIPART from DomainNet (Peng et al., 2019).

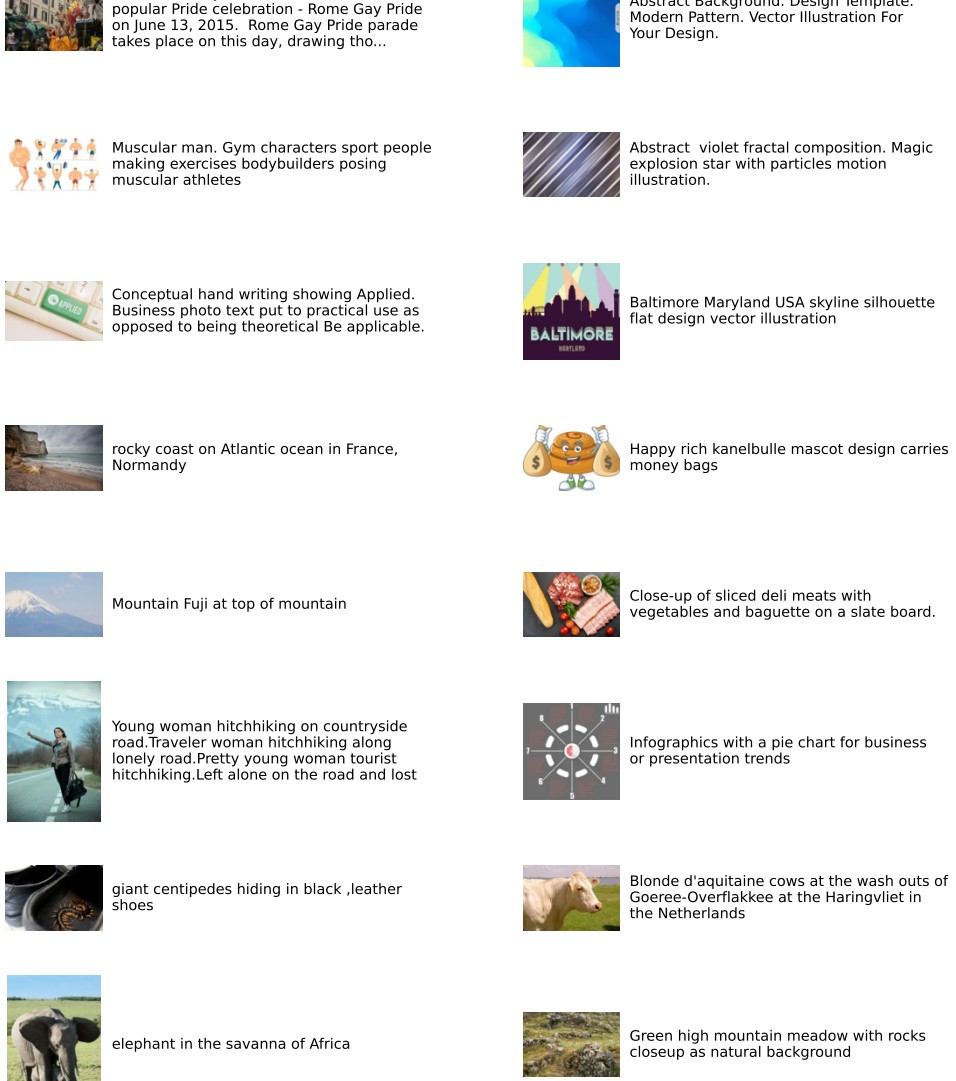

Figure 11: **Random training samples from Shutterstock**

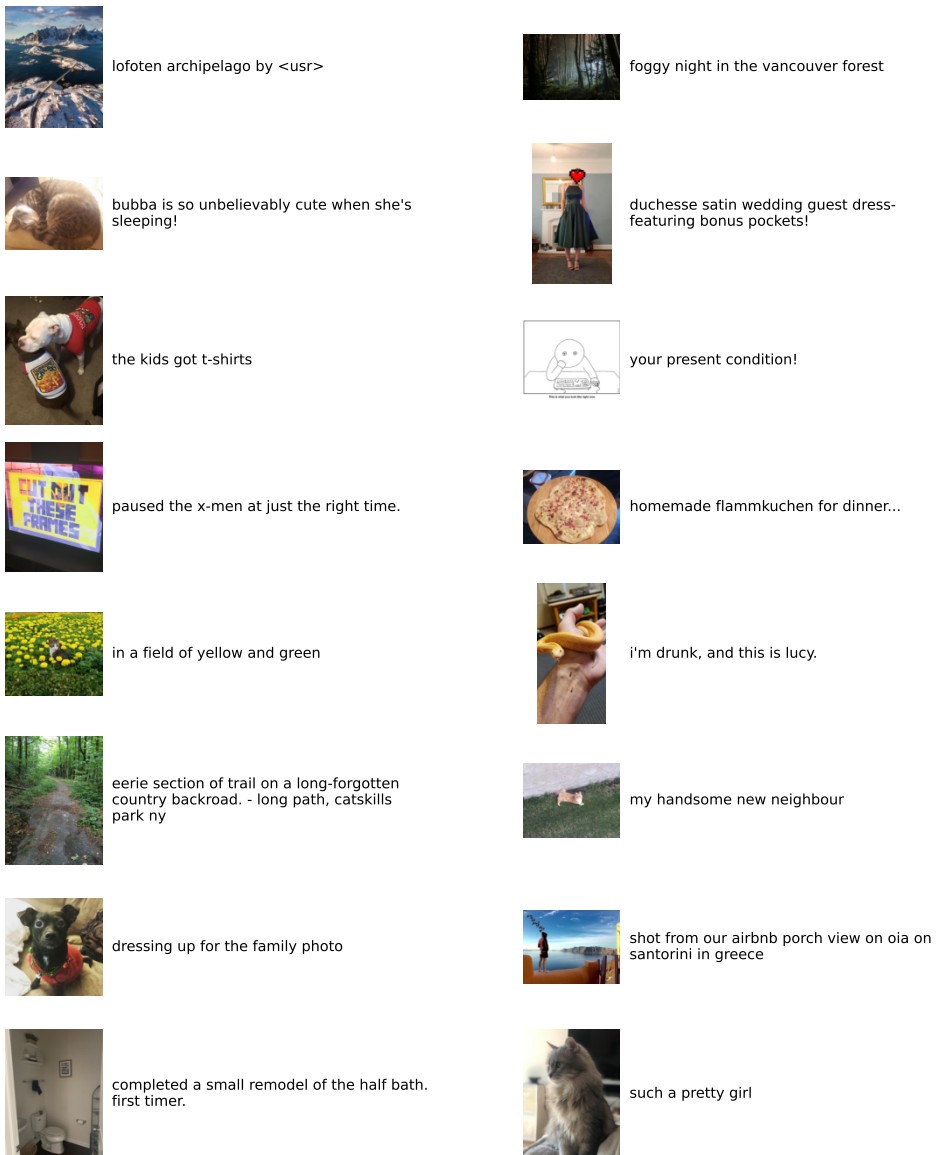

Figure 12: **Random training samples from Redcaps**

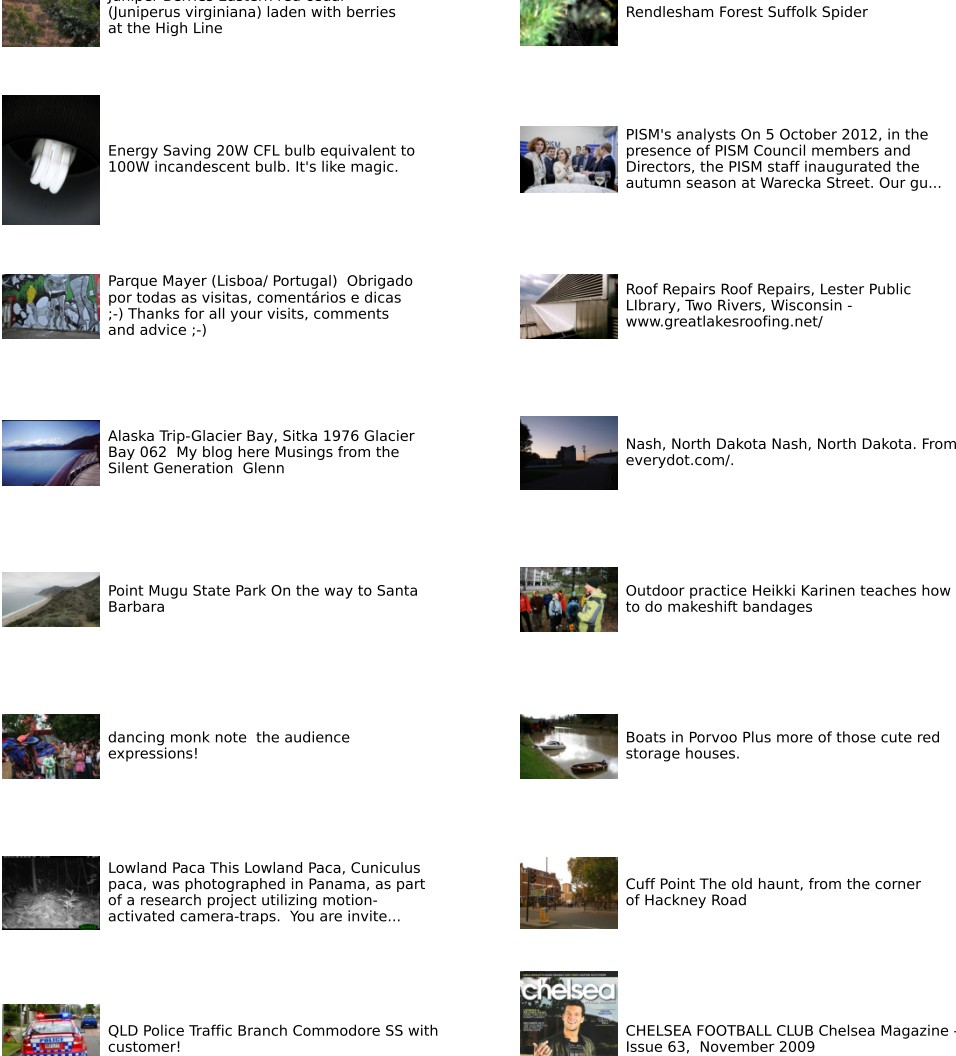

Figure 13: **Random training samples from YFCC**

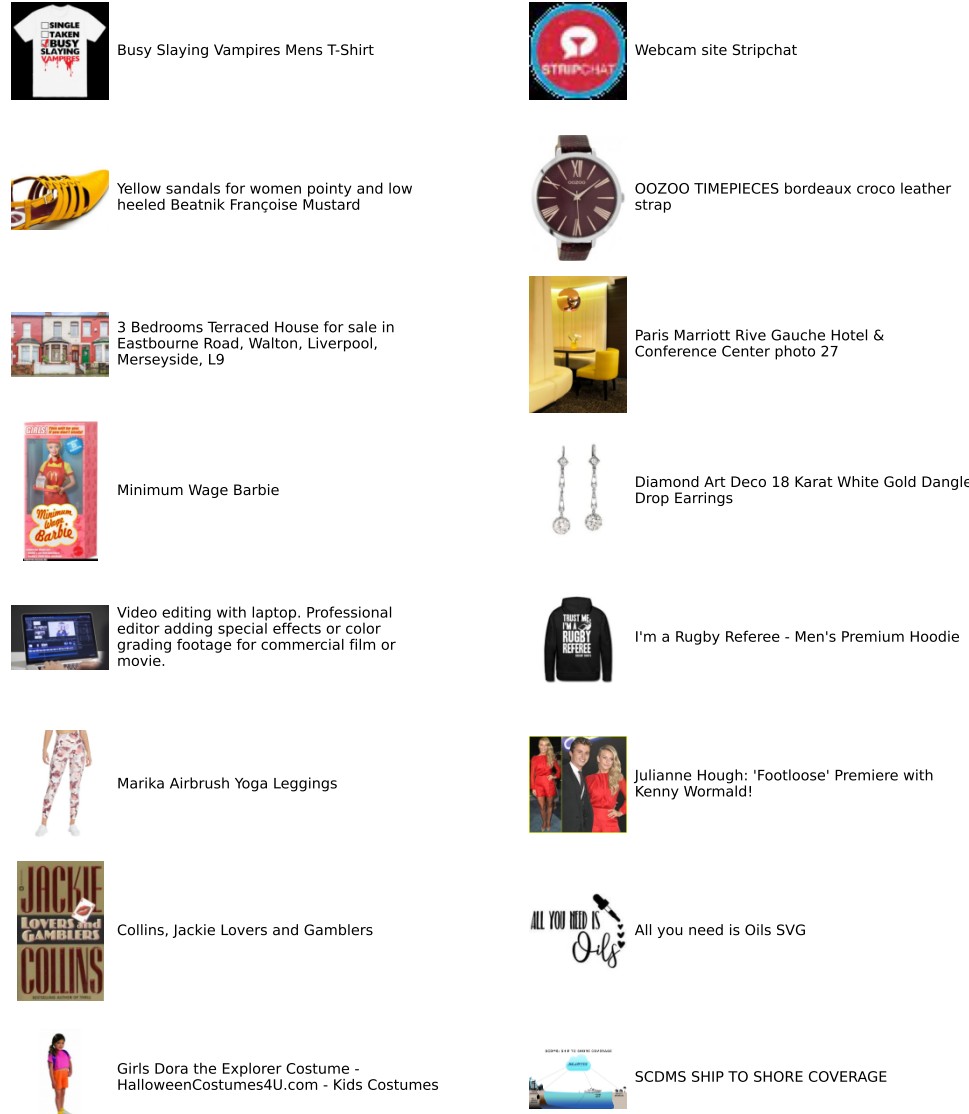

Figure 14: **Random training samples from LAION**

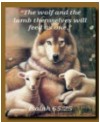 <PERSON> `` The wolf and the lamb shall feed together, and the lion shall eat straw like the bullock: and dust shall be the serpent's meat. They shall not hurt no...

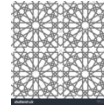 Islamic vector geometric ornaments based on traditional arabic art. Oriental seamless pattern. Muslim mosaic. Turkish, Arabian tile on a white background. Mosque ...

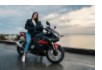 Biker girl in a leather jacket on a black and red color motorcycle

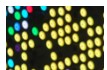 Light Touch Wall digital marketing activation at the Canberra Centre.

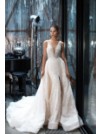 Today's wedding dress inspiration brings us fabulous bridal gowns from creative designer <PERSON>. The Divine Affection lastest bridal collection of <PERSON> wedd...

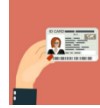 Illustration of hand holding the id card. Vector illustration flat design.

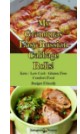 Easy Cabbage Rolls that are <PERSON>, <PERSON> and have no rice! <PERSON> budget friendly comfort food recipe adapted from my Russian grandmother!

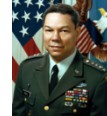 <PERSON>: U. <PERSON> in United States Army. First <PERSON> appointed to that position. First, &, so far, only <PERSON> to serve on Joint Chiefs of Staff. Black H...

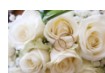 Wedding rings on a bouquet of roses stock photos

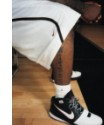 <PERSON> tattoo, the American number 23 from Akron, United States

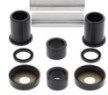 All Balls Swinging Arm Bearing Kit for Yamaha XT225 | XT250 Serow 1993 to 2007

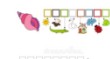 Search the hidden word, the simple educational kid game. stock illustration

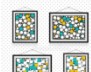 Different types of photo frames with circles and squares on the wall - background template stock illustration

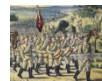 The Russian army entering Prussia, 1914 : News Photo

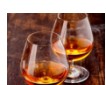 The art of good drinking

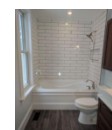 Modern Bathroom Makeovers 20 Design Ideas For a Small Bathroom Remodel. Modern Bathroom Designs On A Budget Minimalist Small Bathrooms, Modern Small Bathrooms, Mo...

Figure 15: **Random training samples from Conceptual Captions**

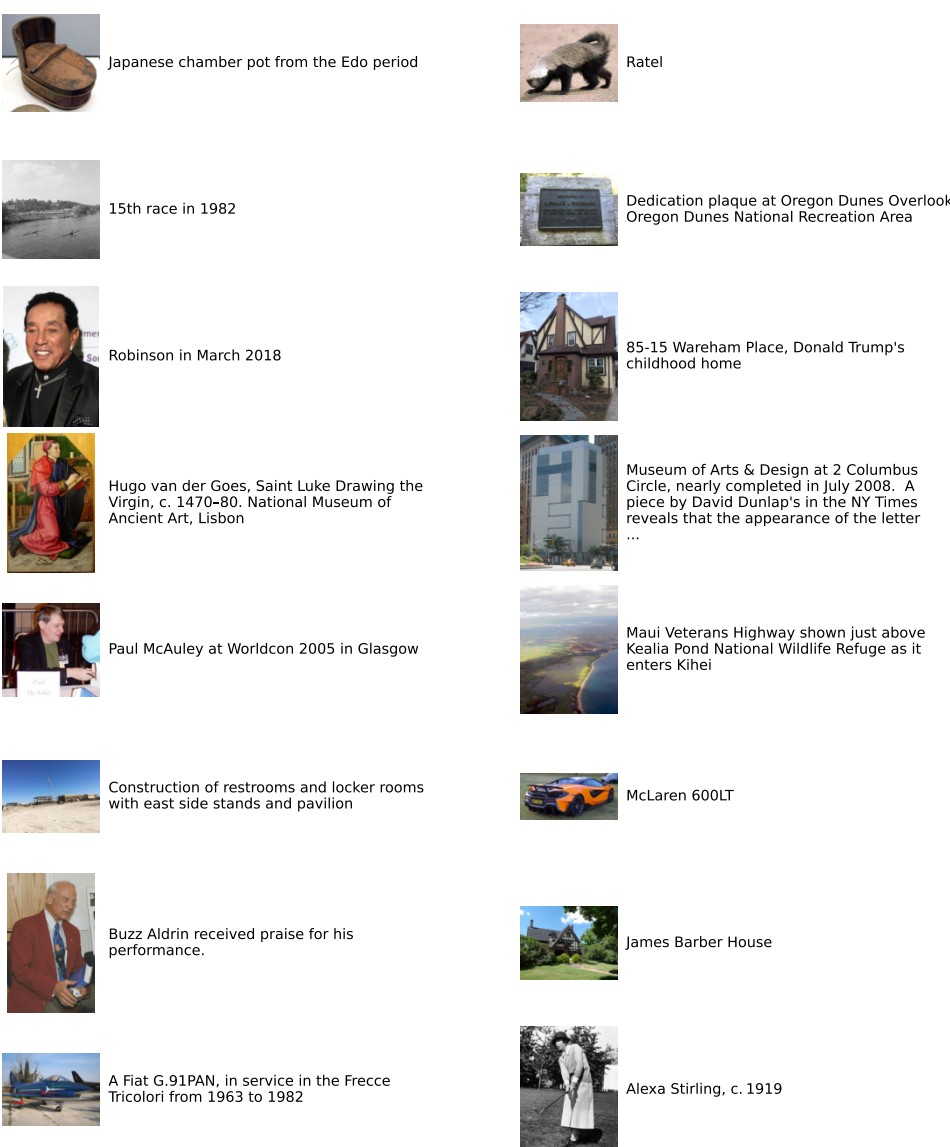

Figure 16: **Random training samples from WIT**

| Downstream Task | Description |
|---|---|
| CIFAR100 | The task consists in classifying natural images (100 classes, with 500 training images each). Some examples include apples, bottles, dinosaurs, and bicycles. The image size is 32x32. |
| DTD | The task consists in classifying images of textural patterns (47 classes, with 120 training images each). Some of the textures are banded, bubbly, meshed, lined, or porous. The image size ranges between 300x300 and 640x640 pixels. |
| CALTECH-101 | The task consists in classifying images of objects (9144 images in 101 classes plus a background clutter class), including animals, airplanes, chairs, or scissors. The image size varies, but it typically ranges from 200-300 pixels per edge. |
| PETS | The task consists in classifying images of cat and dog breeds (7000 images in 37 classes). Images dimensions are typically 200 pixels or larger |
| REAL | The task is a subset of larger DomainNet from six distinct domains, including photos (real), painting, clipart, quickdraw, infograph, and sketch. Total size of 172,000 |
| CLIPART | The task is a subset of larger DomainNet from six distinct domains, including photos (real), painting, clipart, quickdraw, infograph, and sketch. Total size of 172,000 |

Table 3: Details on the downstream datasets used in the experiments.

