# OpenReview forum: "The Role of Pre-training Data in Transfer Learning"
_ICLR.cc/2023/Conference — Submitted to ICLR 2023_

### Official Review · Reviewer_6HTo · 2022-10-24

**Confidence:** 4
**Correctness:** 3
**Technical Novelty And Significance:** 3
**Empirical Novelty And Significance:** 3
**Recommendation:** 6

**Clarity, Quality, Novelty And Reproducibility:**

The paper is clearly written and the quality is fine. There is no significant novelty in this paper.

**Strength And Weaknesses:**

Strength:

The experiment is extensive and adds more pre-training models and datasets compared to previous papers.



Weaknesses:

The paper does experiment with different pre-trained models and datasets in the downtream end, which has already been done in previous papers like [1,2,3]. The paper admits that "we extend these results to more pre-training datasets and methods", which is not a substantial contribution to the community according to my criteria. It is better to propose a novel framework to explain the extensive empirical observation.

[1] Samira Abnar, Mostafa Dehghani, Behnam Neyshabur, and Hanie Sedghi. Exploring the limits of large scale pre-training. arXiv preprint arXiv:2110.02095, 2021

[2] Donghyun Kim, Kaihong Wang, Stan Sclaroff, and Kate Saenko. A broad study of pre-training for domain generalization and adaptation. arXiv preprint arXiv:2203.11819, 2022.

[3] Kornblith, Simon, Jonathon Shlens, and Quoc V. Le. "Do better imagenet models transfer better?." Proceedings of the IEEE/CVF conference on computer vision and pattern recognition. 2019.

The phenomenon of more downstream data leads to less benefit of pre-training is studies in [4]. Could the author give a discussion on the connection and difference between [4] and this paper?

[4] Ziquan Liu, Yi Xu, Yuanhong Xu, Qi Qian, Hao Li, Xiangyang Ji, Antoni B. Chan, Rong Jin, "Improved Fine-Tuning by Better Leveraging Pre-Training Data" Neural Information Processing Systems (NeurIPS), 2022

**Summary Of The Paper:**

The paper investigates the performance of pre-trained models in downstream classification tasks.

**Summary Of The Review:**

I would suggest a weak rejection in this phase.

---

> ### Author Response · Authors · 2022-11-15
> **Response to Reviewer 6HTo**
>
> We thank the reviewer for their valuable feedback on related works. We also appreciate them recognizing the thoroughness of the empirical aspects of the work.
>
>
>
> ## Research questions
> We would like to note that the main focus of this paper is the role of data distribution in transfer learning. We set up a series of extensive yet systematic experiments to answer our research questions as follows, which we believe previous works fell short of:
> - To what extent the pre-training distribution is important as more data is available in the target task (Figure1),
> - How much is expensive labeling worth compared to noisier but larger pre-training data (Figure2),
> - what is the role of scaling pre-training data (Figure3).
> - Finally, we also looked at the effect of the pre-training method (Figure 2, Figure 4, and Figure5).
>
> While we agree with the reviewer, we note that the mentioned literature inspired our related works, **we here highlight the main key difference with the mentioned literature by reviewer:**
>
> 1. Abnar et al and Kornblith et al: The main key difference between these works and ours is the research question: While they try to see what is the relation between upstream and downstream accuracy, our main focus is to see what is the role of data distribution in transfer accuracy. Therefore Kornblith et al include only ImageNet-1K pre-trained models and Abnar et al. studied ImageNet-21K and JFT-300m, mixing the effect of data size and data distribution.
>
>     While they both covered supervised setting with only one modality of vision, we studied image-language and also image-image contrastive learning. Abnar et al. had access to the internal Google dataset of JFT, but our main focus is open-source datasets and therefore covered a wide range of different distributions (2 vs. 7 pre-training datasets). We also compared contrastive image and language with supervised pre-training in figure2. We also plan to extend the scale of our supervised models:  a model pre-trained supervised with ImageNet-21K would be compared with the same distribution but contrastively trained with ImageNet-21K captions (similar to our figure 2: Flickr vs Clip-captions).
>
>     While there is no few-shot study in Kornblith et al, Abnar et al, also looked into the few-shot setting (with linear probe finetuning). In contrast to Abnar et al, we observe that scaling CLIP pre-training to LAION-2B shows non-saturating improvement for CIFAR100. Another minor difference between us and Abnar et al is that they only considered 1000 samples for full finetune (VTAB-highly overlapped with few-shot), while we covered the whole range from 1 shot to all samples on the target task.
>
> 2. Kim et al: We believe that the most similar study to our work is Kim et al (domain generalization), but there are two main key differences: 1) they do not study few-shot transfer (where we see the most impact of pre-training distribution) 2) their main drawback is that they did not provide a set of controlled studies because they are limited to available pre-trained models. For example, their figure 2: When comparing the role of data distribution, top-right to bottom-left (ImageNet-22K vs. JFT-300m), they change the dataset size and also architecture, and the reader wonders if JFT has a better distribution for transfer or the observed effects come from more data/better architecture? We removed this effect by control experiments in Figure2 where we show that ImageNet distribution shows superior performance than LAION.
>     When studying image+language (their figure 2, bottom right), they compare ImageNet-1K supervised with a much larger multimodal dataset, and the reader wonders if the observations are due to the pre-training method (loss), size, or the role of language.
>
>     For each of our research questions, we try to remove all the impacting factors except the main question, e.g. when in Fig1 we study the difference between distributions we fix architecture, size, and pre-training method. When in Fig3, we study the effect of pre-train size, we fix the distribution, architecture, and pre-training method and only vary the size of the dataset.
> 3. Liu et al: We thank the reviewer for bringing this recent related work to our attention.
> While they also observe the diminishing effect of pre-training on transfer learning, we studied the trend of this effect in the range of 1-shot to full data from the target task in Figure 1. In contrast to their conclusion that “training from scratch has the final performance that is no worse than pre-training strategy”, we observe that scaling the number of pre-training samples (Fig1 and Fig3) shows superior performance on training from scratch on studied target tasks. We attribute such a difference to their limited size of pre-training size, where they considered Imagenet-1K (1.2m) vs. our large study of 2.7m, 15m, 400m, and 2B samples of pre-training.

---

> > ### Comment · Reviewer_6HTo · 2022-12-08
> > **response**
> >
> > Thanks for the response. My major concern is addressed in this phase. Though I appreciate the comprehensive empirical study, I still think there should be a clear framework in understanding the effect of pre-training in the downstream.
> >
> > Thus, I will raise my score to 6 and will not object to accepting the paper. It will be appropriated if the discussion about Liu et al is added to the final version.

---

> > > ### Author Response · Authors · 2022-12-08
> > > **Response to Reviewer 6HTo**
> > >
> > > Many thanks for engaging in the discussions about close related works. We would definitely include discussion (e.g. Liu et al) in the final version. As you mentioned we would appreciate it a lot if you can increase your score so that it reflects in the open review system.

---

> > > > ### Comment · Reviewer_6HTo · 2022-12-10
> > > > **Response**
> > > >
> > > > Thanks, the official score is updated.

---

### Official Review · Reviewer_qem9 · 2022-10-27

**Confidence:** 4
**Clarity, Quality, Novelty And Reproducibility:** see above
**Correctness:** 3
**Technical Novelty And Significance:** 2
**Empirical Novelty And Significance:** 3
**Recommendation:** 5

**Strength And Weaknesses:**

The experiments in this paper are extensive, comparing numerous pretraining datasets with different downstream tasks.  This in itself is a useful contribution, particularly if the authors make public the results as a csv file for further analysis, which I would encourage.

However, I think the analysis in this paper and its presentation could be clearer and have more material conclusions.  The largest points (a-c) in my summary above are not particularly informative, as they describe general trends, which at best corroborate known behaviors.  The observation that pre-training choices matter most (quite a lot, in fact) for few-shot settings is perhaps the most interesting one of these.

On the other hand, the paper enumerates many observations describing interesting behaviors in each of the sections --- so some detailed observations are there, but with few conclusions.  For example, at the end of section 4.1, there are 5 enumerated points, which describe many details but few clear conclusions.  Point 2 says "Pre-training a CLIP model on a 2 times larger yet noisy dataset LAION-1m can outperform the IN1K-Flickr-Captions or yield very similar performance".  This merely points to the data and restates in words what the plots show, which is fine if supporting a larger argument, but it is left stranded.  What exact conclusion can be drawn from this?  Does this difference support or elaborate on the larger trends observed in this section?

In addition, most of the figures, while clear, do not always seem the best plots for making the points in the text.  All are organized with number of shots on the x axis, and datasets as point shape/color.  Another view that could work well could be a colored matrix table showing accuracy values as entries, with pretraining datasets on one axis, transfer dataset on the other axis, and transfer training shots as tiled copies of the plot running along the page --- this would make clearer conclusions like 4.1 first sub-section point (2) which says Shutterstock and LAION are best performing, and could also visualize trends in these orderings as transfer shots increase.  There are likely many other figures that could work, and I think more different views could be included to help make more trends apparent.




I also have a few questions on some of the details, enumerated below:


* since SimCLR is better than CLIP pretraining on the three transfer tasks, this immediately brings up the question of whether the rest of the pretraining dataset comparisons might be explored with SimCLR as well.  At the very least, this might be mentioned in the future work section, as the findings are still relevant for CLIP training.

* page 7 sec 4.1 point 4:  should this say "in1k-clip-captions" instead of flickr?

* fig 3, 4, 5:  why not also compare on pets and domainnet?

* sec 4.3:  Why suddenly switch to LP-FT?  Especially if this makes a difference for simclr vs clip, it would be good to compare with FT only as well.  The rest of the paper used this method, so switching methods appears inconsistent.

* sec 4.3 point 2:  Are there any conclusions or hypotheses about why the best-performing dataset is different between clip and simclr?  Perhaps diversity or relevance of images to the downstream tasks, vs of the captions?

* Figure 4:  only SimCLR results are in these figures, so it's difficult to compare between SimCLR and CLIP --- it would be good to have a figure with both included to facilitate this comparison for sec 4.3

* While the pre-training datasets here do differ in the ways described, such as size or noisiness, they have other differences as well --- for example size of the vocab in the captions, diversity of the objects represented in image data.  Do these affect downstream performance as well, and how might those be controlled for or studied?


**Summary Of The Paper:**

This paper investigates the effect of pre-training dataset on transfer learning, when pretraining by text caption contrastive matching using CLIP.  Several different pre-training datasets are compared, using six transfer task datasets (three in later experiments), and in both few-shot and full-data scenarios.  The experiments are extensive, with a fairly rich set of results from which conclusions might be drawn.  The paper enumerates many interesting points in these comparisons, and has a few main takeaways, including: (a) choice of pre-training dataset matters most for low-data scenarios, (b) smaller but cleaner pre-training data can out-perform noisier captions-based data for datasets up to around 15x smaller, (c) more pretraining data helps, but also mostly for low-data transfer tasks.


**Summary Of The Review:**

Overall, this paper presents extensive experiments comparing different pretraining data sources.  This is an interesting set of results, but in my opinion under-utilized in the analysis and presentation.  More exact overall conclusions might be drawn than those like "more data is better" or "cleaner data is better".  The detailed points in each section might be used to help argue for more tangible conclusions, but they aren't clearly stated yet.

---

> ### Author Response · Authors · 2022-11-15
> **Response to Reviewer qem9**
>
> Thanks for your valuable and constructive feedback. We are grateful for recognizing the thoroughness of our experiments. We sincerely hope that you consider increasing your score if your main concern is addressed.
>
> ## Presentation of the contributions
> We have improved the text in the new draft. We specifically changed the flow in the effect of data curation to follow our main research question on the role of data distribution. We believe that we are the first to study the role of data distribution by carefully ablating other factors through a set of well-controlled experiments.
>
> **We address all specific concerns raised by the reviewer as follows:**
>
>     Extensive experiments
> We have included the results for all [4000+ experiments](https://tinyurl.com/yta9e6j4) along with the [code](https://tinyurl.com/25zdy8f4) to reproduce the results. We also plan to publish the trained checkpoints to save the used computational resource (32800 Training Unit-Hours for AI Platform) for future studies.
>
>     Detailed observations but few conclusions
> We have changed the text to reflect the reviewer’s concern. Specifically, we added supervised pretraining on ImageNet to Fig2 and rewrote the data curation in section 4. We believe this section no longer restates in words what the plots show, but rather supports the large argument for the role of pre-training data and answers an important question: “What is the worth of ImageNet expensive labeling?”
>
>     Better visualization
> According to reviewer’s suggestion, we change the presentation of Figure 1 for a better view of exact performance numbers on different data distributions and datasets, depicted in Appendix [fig 7](https://tinyurl.com/455b8wc3).
>
>
>
>     Extension of experiments
> we have extended our experiments to include more downstreams for Fig1, Fig3, and Fig4 (585 more experiments).  Figure1 includes 3 more downstream tasks shown in [fig 6](https://tinyurl.com/jc45z47c). We note that our observation of those figs still holds for the new datasets. Additionally, the extension of fig 3 is shown in Appendix [fig 8](https://tinyurl.com/3k687hbz). See also [fig 10](https://tinyurl.com/3r8an3vy) for an extension of fig 4, where findings in fig4 still hold true for other downstream tasks. We also observe that similar to using CLIP, Redcaps using SimCLR shows superior performance on PETS.
>
>     page 7 sec 4.1 point 4: should this say "in1k-clip-captions" instead of flickr?
> No. As can be seen in the fig2, training on LAION-15m is as good as IN1K-CLIP-captions and better than IN1K-Flickr-captions. We note the difference between IN1K-Flickr-captions and IN1K-CLIP-captions comes from the quality of the captions and also the size of the dataset (1m  vs. 0.5m).
>
>
>     LP-FT
> All other plots for CLIP transfer are based on FT using zero-shot initialization. However, the same approach for SimCLR checkpoints is not possible as SimCLR can not be used zero-shot. Therefore, we use LP-FT for both to enable a fair comparison between SimCLR and CLIP.
>
>
>     Why the best-performing dataset is different between clip and SimCLR?
> We hypothesize that the heavy augmentation throughout the training of SimCLR gives a better representation for SimCLR pre-trained models to transfer on target tasks. We think that pre-training CLIP models with the same augmentation of SimclR would validate this hypothesis but given the computational resources need to check for this, we leave that for future work.
>
>     Fig4 has only SimCLR results
> We note that the main goal of Fig4 is to see if we observe similar trends as Fig1. However, in order to compare SimCLR and CLIP, we included Fig5. As stated in 4.3, in Fig5, we used LP-FT for both CLIP and SimCLR, to have a fair comparison with respect to the initialization of classification layers for target transfer.
>
>
>
>     Differences in pre-training datasets
> We also had a closer look at different pre-training datasets and target tasks and visualized random subsamples of each pretraining dataset in Appendix G.1, figs 11-16. Specifically, when we look at why Redcaps shows superior performance for PETS, we noticed that Redcaps has many samples of animals (see [fig 12](https://tinyurl.com/yudcmu6b)). We also looked at the top 20 words in the captions of pre-training datasets in  Appendix G.1, Table 2: Redcaps include “cats” and “dogs” as most common words. Shutterstock also includes words like “background”, “design”, “pattern”, “and texture” showing high similarity to DTD (Describable Textures Dataset). WIT also mostly represents geo-graphical and people topics, which is far from studied downstream tasks, confirming why WIT shows the worst performance.
> Regarding the diversity of the object for transfer performance, one may look at the representation learned by different pre-training using methods such as CKA (Kornblith, 2019). Regarding captions, one may quantify the difference between the language of different sources by measures like perplexity.

---

> > ### Author Response · Authors · 2022-12-11
> > **Second response to reviewer qem9**
> >
> > This is a kind reminder regarding your last response that you would increase your score. We sincerely appreciate your engagement in the discussions. Thanks to your feedback on the first phase we included more experiments leading to interesting results. We will also include your new suggestion in the final version.

---

### Official Review · Reviewer_thHE · 2022-10-27

**Confidence:** 4
**Correctness:** 3
**Technical Novelty And Significance:** 2
**Empirical Novelty And Significance:** 3
**Recommendation:** 5

**Clarity, Quality, Novelty And Reproducibility:**

The paper is easy to read, with good clarity.
The quality is around the borderline.
The novelty is low.
The reproducibility depends on whether they release the code.


**Strength And Weaknesses:**

# Strength:
A systematic evaluation of the impact of pre-trained data is presented. My educated guess is that this paper requires extensive experiments with massive computation resource. I suggest that authors clearly state or estimate the rough computation resource used for this paper. I suppose curation of those large-scale datasets also contributes largely to the total labor work. In a word, although no novel methods or ideas are presented in this paper, I appreciate the work of verifying common-sense conclusions using large-scale empirical experiments.

Since the authors mention that "our large-scale experiments yield more than 1000 trained networks", all the trained networks can be released if possible, which can be considered as another contribution, just like the paper "Model Zoos: A Dataset of Diverse Populations of Neural Network Models".

# Weaknesses:
The main weaknesses is that conclusions in this paper seem to be known or trivial. It is expected that pre-training dataset is initially important for low-shot transfer but the differences between distributions are diminished as more data is made available for fine-tuning. There are no additional analyses, e.g. on why "Shutterstock and LAION being the best performing pre-training datasets for different downstream tasks, while WIT yields the worst performance in most cases".

In the discussion section, the authors say that "in the future, a sea of pre-trained models will be available for download from the Internet. Therefore, researchers and practitioners will be faced with the question of where to begin." This should be discussed along with the following papers, where they exactly want to address the problem:
LEEP: A New Measure to Evaluate Transferability of Learned Representations, ICML 2020
LogME: Practical Assessment of Pre-trained Models for Transfer Learning, ICML 2021


**Summary Of The Paper:**

Paper 5991 presents an empirical study to address the question of what data and method should be used for pre-training. The scope is limited in the pre-training distribution for image classification. To be specific, the paper investigates the impact of dataset size / pre- training method / whether the dataset is curated on transfer learning. The main take-home message is that pre-training dataset is initially important for low-shot transfer but the differences between distributions are diminished as more data is made available for fine-tuning.

**Summary Of The Review:**

This paper presents some expected conclusions verified by large-scale experiments.


--- Post rebuttal review ---
I have read reviews and responses from authors and peer reviewers. It seems we are all concerned with the novelty of those results. There are so many suggestions in the review to be incorporated into the paper, which I think is beyond the scope of a revision. I encourage the authors to dive into several specific findings rather than presenting many take-away messages without going into them deeply.

---

> ### Author Response · Authors · 2022-11-15
> **Response to Reviewer thHE**
>
> We sincerely thank you for your valuable feedback. We are glad that the reviewer recognizes the thoroughness of our experiments and the significant impact it may have on future studies. Below we address reviewer's comments. Please let us know if there are any remaining questions. We hope that given the reviewer’s positive view of the experimental design and empirical results, they increase the score if they are satisfied with the answers to their questions.
>
> ## Summary of our Conclusions ##
> Below we summarize the investigation performed in each figure and corresponding novel conclusions:
> 1. Figure 1 investigating role of the pre-training data:
> - We identified a pre-train data source, namely Shutterstock, that outperforms other pre-training distributions for most downstream tasks and motivating future work to study their strategy to create the datasets.
> - We identified low-shot vs. high-shot regimes based on transfer saturation of pre-training models for different target tasks.
> - We identified some pretraining datasets that perform better on specific target tasks (Redcaps on PETS and Shutterstock on DTD), and explored why this is happening (Appendix G.1)
>
> 2. Figure 2 and 3 on role of data curation: We observed that pre-training ImageNet-1K is worth of 15x-2000x noisier image and language pairs.
>
> 3. In contrast to Abnar et al. [1] which observed a saturation effect on supervised pre-training and transfer,  Billion size pre-training on transfer still shows increasing performance to CIFAR100 (Fig3).
>
> 4. Figure 5 on the role of pre-training method:
>  We observe that pre-training with SimCLR is showing better performance than CLIP, ablating other factors (Fig5)
>
> **Below we address all specific questions raised by the reviewer.**
>
>     Extensive experiment
> We are thankful for the recognition of our systematic evaluation of our extensive experiments. Thanks to the reviewer’s feedback, we calculate the number of experiments and estimate the total computational resource as follows:
> Number of experiments: 3862
> Computational resource: 32800 Training Unit-Hours for AI Platform ~ 6000 hrs x V100
> We have included the list of all experiments, their results, and the code in the [anonymous link](https://tinyurl.com/25zdy8f4). As suggested by the reviewer we also plan to release the trained network as a model zoo.
>
>     Why some pretraining datasets are performing better on specific target tasks?
> Thanks to this point of the reviewer, we had a closer look on different pre-training datasets and target tasks and visualized random subsamples of each pretraining dataset in Appendix G.1,Figures [11](https://tinyurl.com/bdzn6uzm) -[16](https://tinyurl.com/yckzcrdb). Specifically, when we look at why Redcaps shows superior performance for PETS, we noticed that Redcaps has many samples of animals (see [Figure 12](https://tinyurl.com/yudcmu6b)). We also looked at the top 20 words in the captions of pre-training datasets in  Appendix G.1, Table 2: Redcaps include “cats” and “dogs” as most common words. Shutterstock also includes words like “background”, “design”, “pattern”, “and texture” showing high similarity to DTD (Describable Textures Dataset). WIT also mostly represents geo-graphical and people topics, which is far from studied downstream tasks, confirming why WIT shows the worst performance.
>
>     Relation to LEEP, and LogME:
> While we agree with the reviewer on the closeness to the specified papers, we highlight the following differences:
> (1) their main focus is to develop a measure to predict the full-finetune accuracy without actually fine-tuning on the downstream task. While we also cover full-finetune accuracy in our plots, our main focus in figure 1 is on studying the extent to which pre-training data affect transfer accuracy. Looking at few-shot and full-shot also gives us the ability to study the effect of transfer learning as more target data become available.
> (2) even in the full-finetune scenario, their results on predictability are limited to supervised ImageNet-1K pretraining, while we scale both pre-training distributions and size (Fig3). We also compare the role of language by comparing CLIP and SimCLR in Fig5
> (3) While LEEP only covers MNIST and CIFAR100 for target transfer, LogME extended their results to more downstream tasks and covered self-supervised settings. However, all the target tasks are internet-crawled datasets, while we extended our results to domain-specific datasets too (CCT-20, Cassava Leaf Diseases, and EuroSAT), see Appendix [Figure 6](https://tinyurl.com/jc45z47c).

---

> ### Author Response · Authors · 2022-12-11
> **request from reviewer thHE**
>
> Dear reviewer,
> Given our last discussion and the current update from other reviewers, we kindly ask you to keep your previous score or consider increasing your score.
>
> Best regards,
> Authors

---

### Official Review · Reviewer_bvES · 2022-10-28

**Confidence:** 4
**Clarity, Quality, Novelty And Reproducibility:** The paper is easy to read. The result…
**Correctness:** 3
**Technical Novelty And Significance:** 1
**Empirical Novelty And Significance:** 1
**Recommendation:** 3

**Strength And Weaknesses:**

Strength
- The paper empirically studies the effects of pre-training (data source, pre-training methods, loss function) on downstream tasks.
- The paper provides fine-tuning results with different pre-training methods (e.g., CLIP based ResNet-50 model), which might be new for practitioners.

Weakness
- The questions asked in the paper were already well studied and the answer is not surprising. The major take-away conclusion of “the effect of pre-training diminishes as more training data are available” is a well known fact and was shown in literature on transfer learning and model selection such as [1].
- The downstream tasks are quite limited (only 6) and some of them have quite saturated performance (such as PETS). It is not quite convincing to draw conclusions with these simple datasets.
- As identified by the authors, there is no explanation or insights about why certain pre-trained models work better than others.
- Only a single architecture (ResNet-50) is studied. Not clear whether the findings generalize to other architectures as well.
- [minor] The authors claimed the best performance on the test was picked with variations of learning rates and batch size, however, the appendix said only a single batch size is used.

[1] Deshpande et al, A linearized framework and a new benchmark for model selection for fine-tuning, 2021
[2] Bolya et al, Scalable Diverse Model Selection for Accessible Transfer Learning, 2021


**Summary Of The Paper:**

This paper studies the effect of the pre-training on transfer learning for image classification. Their study covers 7 pre-training datasets, 6 fine-tuning datasets, 2 pre-training methods for both few-shot and full finetuning. They have made several empirical observations. They find that the pre-training dataset is initially important for low-shot transfer and the importance of a good pre-trained model diminishes when fine-tuning on more data.


**Summary Of The Review:**

This paper provides empirical observations on the effectiveness of pre-training. However, not much new insights are provided.

---

> ### Author Response · Authors · 2022-11-15
> **Response to Reviewer bvES**
>
> We thank the reviewer for their detailed feedback. Below, we first address their main concern about experimental results and then try to designate the issues in detail. We sincerely hope that they consider increasing their score if their main concern is addressed.
>
> ## Main contribution
> Here we address the reviewer’s main concern regarding the novelty of the work. We note that the main question of this paper is the role of data distribution, which we believe related works fall short of answering.  All our systematic experiments focus on data while carefully ablating the other impacting factors. We believe that our research questions (detailed below) have not been investigated before.  One concrete example of our novelty is the role of data curation in Fig 2: We asked what is the real worth of expensive labeling compared to noisier but larger data.
>
> **We address all specific concerns raised by the reviewer in detail as follows**
>
>     Related works and our research questions
> While we thank the reviewer for bringing [1, 2] to our attention, we summarize our difference to the mentioned related works as follows: The main focus of these studies is on developing metrics for predicting the transferability of a model. While we also look at the gap between different pre-training models given a target task.  Our main focus is on the role of data, hence varying data distributions is the main key difference to related works. Through a set of controlled experiments, we answered:
> 1. To what extent the pre-training distribution is important as more data is available in the target task (Figure1),
> 2. How much is expensive labeling worth compared to noisier but larger pre-training data (Figure2),
> 3. What is the role of scaling pre-training data (Figure3).
> 4. Finally, we also looked at the effect of the pre-training method (Figure 2, Figure 4, and Figure5).
>
> We also note that previous works including those mentioned by the reviewer are mostly limited to the supervised setting and only one pre-training distribution, ImageNet-1K.
>
>     Limited downstream tasks
> We extended our results to 3 more downstream datasets in the new draft (Appendix, [Figure 6](https://tinyurl.com/jc45z47c)) While the first six datasets are internet-crawled datasets, these new downstream datasets (CameraTraps, Cassava Leaf Disease, EuroSAT) are domain-specific, and from Kaggle competitions.  Fig 6 shows that our observation in Section 4.1 still holds, i.e changing the pre-training dataset leads to noticeable differences in few-shot transfer performance. In contrast to Figure1, few-shot transfer for newly added datasets shows higher performance, which we attribute to the similarity of the pre-trained models and target tasks.  We also notice that within each new target task, the order of downstream performance for different shots is more intertwined.
>
>     Insights about why certain pre-trained models work better than others
> Thanks to this point of the reviewer, we had a closer look and visualized random subsamples of each pretraining dataset in Appendix G.1, Figures [11](https://tinyurl.com/bdzn6uzm) -[16](https://tinyurl.com/yckzcrdb). Specifically, Redcaps has many samples of animals, explaining why it shows superior performance on PETS. We also looked at the top 20 words in the captions of all pre-training datasets in Appendix G.1, Table 2: Shutterstock includes words like “background”, “design”, “pattern”, and “texture” showing high similarity to DTD (Describable Textures Dataset). WIT also mostly represents geo-graphical and people, which is far from our downstream tasks, confirming why it shows poor performance.
>
>     Extension to other architectures
>
> We note that we ran around 4000 experiments in our study for ResNet-50 and the computation totals to 32000 Training Unit-Hours (~6000 GPU hours on V100s). However, given the limited time and computation at hand, we ran the following experiment to address the reviewer's concern in Figure9 in Appendix. We use ViT instead of ResNet-50 for LAION400m vs. OpenAI400m. While similar to Figure1, the difference between the fine-tune performance is minimal, in contrast, we observe that both models perform very similarly in the few-shot setting. We hypothesize that this observation could be attributed to the similarity between LAION and OpenAI distributions rather than employing a transformer instead of ResNet-50. A controlled study may include replicating Figure1, but with ViT. Because this experiment is computationally intensive on the pre-training side, we leave it for future work.
>
>     Hyperparameter search
> Thanks for bringing this to our attention. We corrected this mistake and now include the list of all experiments for different hyperparameters and their results in a [CSV file](https://tinyurl.com/yta9e6j4) along with the [released code](https://tinyurl.com/25zdy8f4). We also plan to release all checkpoints for the community to reduce the carbon footprint for further studies of transfer learning.

---

> > ### Comment · Reviewer_bvES · 2022-12-09
> > **Thanks for the response**
> >
> > I appreciate the authors' efforts for responding to my comments. However, it does not address the concerns for the novelty. The response to the question and the new summarization just confirmed that the major observations are already well known facts in the model selection or transfer learning literature: 1) the effect of pre-training datasets matters more for few-shot dataset 2) the downstream task benefit more from similar pre-training datasets or "changing the pre-training dataset leads to noticeable differences in few-shot transfer performance" 3) The explanation of why certain model works better is relevant, however, similar explanation or analysis based on domain similarity also exists such as in [3]. Therefore, claiming them as new contrition makes the statement weak.
> >
> > [3] Large scale fine-grained categorization and domain-specific transfer learning, CVPR 2018

---

> > > ### Author Response · Authors · 2022-12-09
> > > **Second response to reviewer bvES**
> > >
> > > We thank the reviewer for involving in the discussion.  Let us disagree with “the major observations are already well-known facts”. We would like to refer you to our 5 research questions in the paper (summarized in the takeaway message response above). Through careful yet extensive experiments we made **13 conclusions** for these 5 questions.  For example, we designed careful step-by-step ablations to see“ How much expensive labeling is worth?”, and conclusions seem very untrivial to us (see un update from figure2 [here](https://github.com/AnonymousMLSubmission/DataDistributionTransfer/blob/main/plots/Fig2-1.png) and [here](https://github.com/AnonymousMLSubmission/DataDistributionTransfer/blob/main/plots/Fig2-2.png) ). We also looked into the difference between contrastive (image only vs. imge+language) and supervised pre-training in transfer learning. We are not aware of such studies in the literature.
> > > While some of the conclusions seem familiar (as you pointed out) we believe that no related works looked into the role of different data distributions, instead at best they compared pre-training to training from scratch and on a small scale (iNaturalist and ImageNet1K in [3]). We believe that even "expected" results need to be validated scientifically, as they are often easily taken for granted without careful analyses.
> > >
> > > In addition, we believe that in the near future we would see many more large-scale datasets for pre-training and therefore it would be crucial to study the effect of different data distributions over gigantic yet noisy datasets. We hope that such studies help to make better datasets in the near future. We sincerely hope that they consider increasing your score if you find some of our questions and conclusions are worth of novelty.

---

### Comment · Area_Chair_AAet · 2022-11-20
**Please update your reviews**

Please make sure that your reviews acknowledge authors’ responses and reflect your current evaluation of the paper. This is particularly important if you didn’t directly engage with the authors during the discussion phase (so the authors don’t know if their response changed your evaluation) or if you expressed an intention to update your rating but did not do so.

Cheers,
AC

---

### Author Response · Authors · 2022-11-24
**Take away messages**

We would like to thank all the reviewers for their valuable feedback and for recognizing the impact of our study on future works. We have changed the draft with respect to reviewers’ concerns. We would appreciate increasing your score if your concerns are addressed and we are very open to discussions. We would like to highlight our main research questions and non-trivial conclusions as follows:

---

# To what extent the pre-training distribution is important for transfer learning? (Figure1)

> **1.** Different pre-training distributions lead to noticeable differences in downstream transfer performance only in low-shot.

> **2.** We identified low-shot vs. high-shot regimes based on the transfer saturation of pre-training models for different target tasks.

> **3.** We identified some pretraining datasets that perform better on specific target tasks (Redcaps on PETS and Shutterstock on DTD) and explored why this is happening based on visual and caption similarities (Appendix G.1)

> **4.** We identified a pre-train data source, namely Shutterstock, that outperforms other pre-training distributions for most downstream tasks, motivating future work to study the dataset peculiarities in more detail.
---

# How much expensive labeling is worth? Which one is better: Supervised or Contrastive?(Figure2 and Figure3)

> **1.** Contrastive pre-training (LAION) needs 15x more data to outperform supervised pre-training (ImageNet-1K), on CIFAR100.

> **2.** Contrastive pre-training (LAION) needs 2000x more data to match/outperform supervised pre-training (ImageNet-1K), on DTD, REAL, and CLIPART. Even 2000x more data from LAION is not enough to math ImageNet pretraining for PETS and CALTECH101.
---


# What is special about the ImageNet distribution? Contrastive pre-training, comparing ImageNet distribution with LAION. (Figure2)
> **1.** Pre-training on ImageNet (with Flickr captions) outperforms LAION with the same size (0.5m samples)

> **2.** We need 2x LAION samples to outperform ImageNet (Flickr captions).

> **3.** We need 15x LAION samples to outperform ImageNet (high-quality CLIP captions).

---

# Pretraining size and saturation (Figure3)
> **1.** Increasing pre-training size has different saturation effects on different downstream tasks. While for CIFAR100, LAION-2B shows a large gap to smaller sizes, even 100x more data of LAION-2B performs similarly to pretraining on LAION-15m for PETS and CALTECH101.

> **2.** LAION shows better performance than YFCC on transfer to studied downstream tasks, both at 2.7m and 15m sizes.
---

# Pre-training methods (Figure4 and Figure5)
> **1.** Different pre-training datasets also lead to noticeable differences in downstream transfer performance in few-shot, when using image-image contrastive pre-training (Figure 4).

> **2.** In addition to the comparison between CLIP and Supervised (Figure2), SimCLR pretraining shows better performance than CLIP, while ablating other factors such as size, and architecture (Figure5).
---

---

> ### Comment · Reviewer_qem9 · 2022-12-04
> **responses**
>
> Thanks for these responses and revisions.  The new experiments including standard ImageNet-1k are in particular an improvement.
>
> I think the points you make in your comment above ("Take away messages") are succinct and quantitative enough to be interesting conclusions.  However, even with the new revisions, the paper itself doesn't have anything so concise as this.  I would suggest putting a bullet list like the above --- almost even a copy/paste of these exact points --- into the paper, perhaps towards the end in a new conclusions section.
>
> I've kept my score unchanged at the moment, but would be inclined to raise to 6 with these more concise takeaways included, as I think those provide interesting enough conclusions when considered in conjunction with release of the experiments results.

---

> > ### Author Response · Authors · 2022-12-05
> > **Second response to Reviewer qem9**
> >
> > We sincerely appreciate your engagement in the discussions. Thanks to your feedback on the first phase we included more experiments leading to interesting results.  We will also include your new suggestion in the final version. Given what you mentioned in the last response we would be very thankful if you increase your score.

---

### Decision · Program_Chairs · 2023-01-20

**Decision:**

Reject

**Justification For Why Not Higher Score:**

Majority of reviewers have the same conclusion that this paper is below the acceptance threshold. There is one review increased score due to the authors' rebuttal efforts. However, some major parts remained unsolved.

**Justification For Why Not Lower Score:**

NA

**Metareview: Summary, Strengths And Weaknesses:**

This paper empirically studies the effects (pre-training data, training losses, low/high-shot finetuning) in transfer learning for the pre-trained models on different downstream computer vision datasets. They found that pre-training is important to low-shot downstream tasks, and the importance diminishes in high-shot downstream tasks. The main concerns of reviews are that the authors do not clearly present their insights, and the novelty is borderline. I suggest the authors spend more time on their manuscript to further separate their contributions from the prior work.

**Summary Of Ac-Reviewer Meeting:**

This paper provides empirical studies on the effectiveness of pre-training.  Most reviewers comment that there are not many new insights in this work, and rates blow the acceptance borderline. The authors tried their best to make responses and reclaim their insights actively. I believe the authors make great efforts in time and resources for their work, and  I strongly encourage the authors to try another publishing track.